# Determinants of chromosome-specific telomere lengths among 2573 All of Us participants

Niyati Jain[1,2], Jiajun Luo[3], Yuqing Yang [4], Kathryn Demanelis[5,6], Habibul Ahsan[1,4,7], Briseis Aschebrook-Kilfoy[4,7], Lin S. Chen[1] & Brandon L. Pierce[1,2,7,8] ✉

Telomere length is a biomarker of aging and disease risk. Most human studies have assessed average telomere length, limiting our understanding of variability across chromosome arms. Using long-read sequencing data from >2500 All of Us participants, we estimate chromosome-specific telomere lengths and characterize sources of biological and technical variation. Telomere length varies by chromosome arm, accounting for 9.1% of total variance. Substantial variance (8.9%) in chromosome-specific telomere lengths is attributable to individual, independent of age, suggesting that inter-individual differences in length are established at birth and maintained through life. Age is inversely associated with length for all arms, but longer arms show stronger association. We demonstrate that chromosome-specific telomere estimates enable analysis of disease associations for individual telomeres and for individuals' shortest telomere. Overall, this work highlights the utility of long-read sequencing for population-scale analysis of chromosome-specific telomere lengths and provides a framework to guide future studies.

Telomeres are DNA-protein structures at the ends of chromosomes that protect against degradation and fusion. As cells divide, the repetitive DNA sequence of telomeres (TTAGGG) progressively shortens, eventually leading to cell senescence[1]. In most human tissues, telomeres shorten over time and are considered a hallmark of aging[2,3].

Telomere length (TL) varies across individuals and cell types, and is influenced by genetic, environmental, and lifestyle factors. The heritability of TL likely has two sources: inherited genetic variation—such as SNPs that influence telomere maintenance[4–9]—and the direct influence of the TL of parental germ cells on the TL of embryonic cells of the offspring[10]. Environmental and lifestyle factors, including pollution, stress, and cigarette smoking, have also been shown to impact TL[11–15]. TL has been linked to disease risk, with shorter TL correlating

with a higher risk of several aging-related diseases[16,17], and longer TL linked to increased risk of certain cancers[18,19]. Studies in yeast[20–22], mice[23], and humans[24–26] have suggested that the shortest telomere or the abundance of short telomeres in cells may be relevant biomarkers for cell dysfunction and disease risk.

Most epidemiological studies of TL have used methods that measure the average TL across all chromosome arms, such as quantitative PCR (qPCR)[27,28], Southern blot[29], or Luminex[30,31]. More recently, bioinformatics tools like TelSeq[32], Computel[33], and Telomerecat[34] have been developed to estimate average TL from short-read whole genome sequencing (srWGS) data. However, TL can also be measured at individual chromosome arms (with the diploid human genome containing 92 arms). Traditional lower-throughput methods include metaphase

[1]Department of Public Health Sciences, University of Chicago, Chicago, IL, USA. [2]Committee on Genetics, Genomics, Systems Biology, University of Chicago, Chicago, IL, USA. [3]Department of Public Health and Medicinal Administration, Faculty of Health Sciences, University of Macau, SAR Macau, China. [4]Institute for Population and Precision Health, University of Chicago, Chicago, IL, USA. [5]Department of Medicine, University of Pittsburgh, Pittsburgh, PA, USA. [6]UPMC Hillman Cancer Center, Pittsburgh, PA, USA. [7]University of Chicago Comprehensive Cancer Center, University of Chicago, Chicago, IL, USA. [8]Department of Human Genetics, University of Chicago, Chicago, IL, USA. ✉e-mail: brandonpierce@uchicago.edu

quantitative fluorescent in situ hybridization (Q-FISH)[35], optical imaging[36–38], single telomere length analysis (STELA)[35], and telomere shortest length assay (TeSLA)[39]. Despite their utility, labor-intensive protocols, specialized sample requirements, and high costs restrict their scalability for large population studies.

Long-read WGS (lrWGS) platforms, such as Oxford Nanopore Technologies (ONT) or Pacific Biosciences (PacBio), provide new opportunities for chromosome-specific TL (csTL) measurement. Long reads, 10−30 kilobases (kb) in length, can span telomeres and their adjacent regions (telomere variant repeat and subtelomere region). The software edgeCase initially demonstrated that hierarchical clustering of long reads based on their sequence can distinguish individual telomeres[40]. Subsequently, Telogator[41] and then Telogator2[42], were developed to estimate TL at each chromosome arm using lrWGS data. This method applies hierarchical clustering and aligns read clusters to subtelomere sequences based on telomere-to-telomere (T2T) reference genomes. Assays that capture and sequence telomere regions, using either ONT or PacBio, have also been developed, including telomere profiling[43], Telo-seq[44], and telobaits[45]. These targeted approaches reduce sequencing costs and increase the feasibility of csTL measurement for large-scale human studies.

These technologies are now being applied to collections of human samples. Karimian et al.[43] applied ONT-based telomere profiling to 147 samples, including 132 healthy adults and 15 individuals with idiopathic pulmonary fibrosis. The study reported variability in TL across arms, with certain arms tending to be shorter (e.g., 17p, 20q, 12p) or longer (e.g., 4q, 12q, 3p) on average. Notably, these differences across arms were evident in cord-blood and adult samples, suggesting they are established at birth and maintained over time, aligning with Q-FISH findings[43,46–48]. Additional research in human populations is needed to improve our understanding of csTL variability, its determinants, and its role in aging and disease.

In this study, we generate csTL estimates for 2573 individuals of diverse ancestry using lrWGS data from the NIH All of Us Research Program (AoU)[49]. We evaluate (i) the performance of Telogator2 by assessing how sequencing coverage, read length, and different sequencing platforms impact csTL estimates; (ii) the distribution and differences in TL across chromosome arms; (iii) the factors explaining variation in TL—including chromosome arm, age, BMI, smoking, ancestry, and individual differences; and (iv) the associations between TL and health-related trait across chromosome arms, as well as with mean TL and shortest TL per individual. This work provides a deeper understanding of TL variability across individuals and chromosome arms, offering insights into its role in aging and disease.

## Results

### Characteristics of All of Us participants

We used Telogator2 to align long reads to chromosome-specific subtelomere sequences and estimate csTLs for 2573 AoU participants (Fig. 1). The AoU lrWGS data was generated at five different sequencing centers: Baylor College of Medicine (BCM), Broad Institute (BI), Johns Hopkins University (JHU), University of Washington (UW), and HudsonAlpha Institute (HA), utilizing three sequencing platforms (ONT, PacBio Revio, PacBio Sequel IIe) with two minimum coverage thresholds (>8x or >12x for mid-pass and >25x for high-pass). These technical differences defined 11 unique batches, with batch sizes ranging from 46 participants (center: HA, platform: Revio, coverage: mid-pass) to 971 participants (center: HA, platform: Sequel IIe, coverage: mid-pass). Among all participants with lrWGS data, 64.8% self-reported female at birth and 64.4% were classified as never smokers (Table 1). Participant ages ranged from 17-90 years (Supplementary Fig. 1). Across all batches, 50% of participants were of African (AFR) ancestry, 28% were of Latino/Admixed American (AMR) ancestry, 13% were of European (EUR) ancestry, and the remaining 9%, classified as other, were of South Asian (SAS), East Asian (EAS), or Middle Eastern (MID) ancestry.

### Long-read sequencing metrics and Telogator2 performance

To understand how technical variation in lrWGS data generation affects csTL estimation, we assessed differences in several key metrics, including read lengths, the number of reads supporting csTL estimation, and the number of chromosome arms with TL estimated, across the 10 batches. While previous analyses have explored how coverage affects the number of csTL estimates generated by Telogator2, its impact on the csTL values themselves remains largely uncharacterized[42]. Additionally, the effects of read length and platform-specific characteristics on csTL estimation have not been systematically studied.

The mean read lengths across batches ranged from ~16 to ~30 kb. All batches produced mean read lengths around 20 kb, except one ONT batch (JHU_ONT_high), which had a higher mean read length of approximately 30 kb (Fig. 2a). Though we expected read length to not greatly influence csTL estimation—since reads of ~20 kb are generally sufficient to capture the full length of most telomeres—we observed moderate correlations between read length and csTL for the JHU_ONT_high (Pearson $r = 0.24$, $P = 5 \times 10^{-48}$) and BCM_ONT_high (Pearson $r = 0.28$, $P = 2 \times 10^{-72}$) batches. This trend was much less apparent in the PacBio batches (Supplementary Fig. 2), suggesting potential platform-specific technical differences.

The high-pass batches showed a higher number of chromosome arms with csTL estimates compared to mid-pass batches (Fig. 2b). However, there was substantial variability in the number of arms with csTL estimates within both high-pass and mid-pass batches, with mean counts ranging from 28 to 81 arms with csTL estimates in high-pass batches and 13−35 in mid-pass batches.

Next, we compared telomeric coverage, measured as the number of reads supporting TL estimation, to the number of chromosome arms with csTL estimates generated by Telogator2. As expected, higher coverage at telomeres was associated with greater number of arms with csTL estimated (Fig. 2c). Notably, high WGS coverage did not always translate to high telomeric coverage, highlighting the importance of assessing telomeric coverage directly when evaluating Telogator2 performance (Supplementary Fig. 3). Additionally, we observed that for a given arm, majority of individuals had only a single csTL estimate (Supplementary Fig. 4), thus, motivating our approach of reporting the average csTL when 2 or more estimates were available or the single estimate when only one was present. Among the batches evaluated, the Sequel IIe high-pass batches showed the highest telomeric coverage and the greatest number of chromosome arms with csTL estimates per sample.

To further assess the effect of sequencing coverage on Telogator2's ability to generate csTL estimates, we leveraged 36 high-coverage (>30x telomere coverage) Human Pangenome Reference Consortium (HPRC) samples[50] and down-sampled the reads to 25x, 20x, 15x, 10x, and 5x telomeric coverage. Consistent with AoU results, increasing coverage resulted in more arms with csTL estimated (Supplementary Fig. 5A). The mean absolute estimation error (calculated as the average absolute difference between csTL estimates from high-coverage and down-sampled coverage) was ~100 base pairs (bp) at 25x coverage and increased to ~1500 bp at 5x coverage (Supplementary Fig. 5B). However, the correlation between csTL estimates present in both down-sampled and high-coverage data remained high across all comparisons (Pearson $r > 0.75$, $P < 2 \times 10^{-16}$). Because lower coverage datasets produce fewer csTL estimates, the correlations estimated for low coverage datasets are based on few observations (Supplementary Fig. 5C−G). This suggests that, although lower coverage increases csTL estimation error, the relative patterns across chromosome arms—such as the ranking of longest and shortest csTL estimates—are preserved, supporting the validity of estimating csTL even for lower-coverage samples.

Lastly, we evaluated Telogator2's performance across individual chromosome arms and found that csTL estimates were frequently

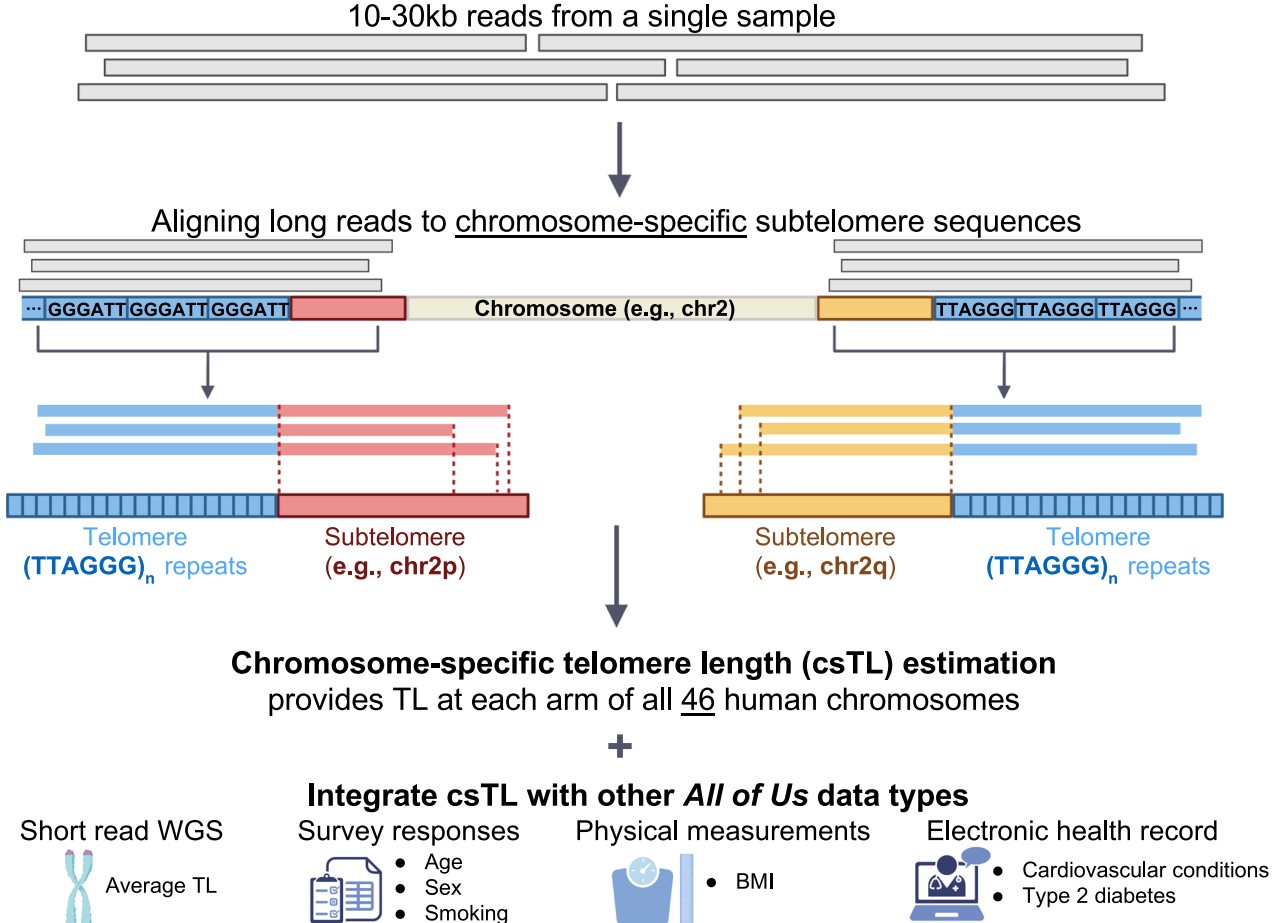

**Fig. 1 | Overview of data types and workflow for chromosome-specific telomere length estimation using long-read sequencing data from the All of Us cohort.** The figure was created in BioRender. Jain, N. (2026) https://www.biorender.com/u4ywlbu.

missing for several arms, even in high-coverage datasets, with >30% of samples lacking an estimate for some (Fig. 2d). The pseudo-homologous regions at the short arms of acrocentric chromosomes (13p, 14p, 15p, 21p, 22p) and the pseudo-autosomal regions at the ends of the X and Y chromosomes facilitate frequent recombination between non-homologous chromosomes[51–53]. Such recombination is known to prevent reliable and unambiguous assignment of telomeres to these arms, as demonstrated by Karimian et al.[43]. The authors found that no reads from these arms in the Pangenome dataset mapped to the same chromosome end across three haploid reference assemblies. Therefore, we excluded these arms from further TL analyses. This exclusion resulted in the loss of one participant, leaving 2572 participants for downstream analyses. Consistent with prior findings, the short arms of acrocentric autosomes (e.g., 13p, 14p, 15p) and the arms of sex chromosomes (e.g., Xp, Yp) were among the underperforming arms across all 5 AoU aggregated batches (defined by platform and coverage) and in the 36 high-coverage (>30×) HPRC samples sequenced using PacBio Revio (Fig. 2d).

**Benchmarking csTLs against TelSeq-derived average TL**
The distribution of csTL estimates across AoU lrWGS batches is shown in Fig. 3a. ONT high-pass batches had systematically shorter csTL estimates across all samples (mean TL < 3 kb for both batches). In contrast, the PacBio batches displayed consistently longer csTL estimates, with mean csTLs ranging from ~3.5 to 5 kb.

To further validate csTL estimates generated by Telogator2, we compared them with estimates from TelSeq, a widely used tool for estimating average TL from srWGS data. To account for technical sources of variability in TelSeq-derived TL, we regressed out batch principal components (PCs) derived from sequencing depth data as described previously[6,8,54], optimally selecting the number of PCs ($n = 30$) that maximized the age correlation. In analyses including all participants, adjusted TelSeq-TL was positively correlated with csTL across all chromosome arms ($P < 0.01$) (Fig. 3b).

Across the five aggregated sequencing batches, the Sequel IIe and Revio batches showed clear positive correlations with adjusted TelSeq-TL across nearly all chromosome arms ($P < 0.01$). In contrast, ONT-based csTL estimates showed the least consistent correlation with adjusted TelSeq-TL, with most chromosome arms showing no significant association ($P > 0.05$) (Fig. 3c). Restricting analyses to non-ONT batches resulted in robust positive correlations between adjusted TelSeq-TL and csTL across all chromosome arms (Supplementary Fig. 6). Given known limitations of ONT data processed with older base callers (e.g., Guppy used in AoU) for estimating csTLs using Telogator2, we excluded ONT-derived batches from downstream analyses to avoid introducing bias. Moreover, the weaker correlations observed for ONT data relative to the smaller PacBio batches further highlights the potential limited reliability of Telogator2 csTL estimates derived from ONT data in AoU (see "Discussion").

**Table 1 | Characteristics of All of Us participants included in csTL analysis using lrWGS data**

| Minimum coverage | High-pass (>25x) | | | | | | | Mid-pass (>12x or >8x*) | | | | All participants (n = 2573) |
|---|---|---|---|---|---|---|---|---|---|---|---|---|
| Platform (n) | Sequel IIe (239) | | | Revio (152) | | ONT (261) | | Sequel IIe (1152) | | Revio (769) | | |
| Center (n) | BCM (71) | BI (77) | UW (91) | BCM (101) | UW (51) | BCM (146) | JHU (115) | BI (181) | HA* (971) | BI (723) | HA (46) | |
| Age, yrs (SD) | 43.5 | 50.8 | 47.0 | 42.9 | 45.2 | 46.0 | 44.2 | 46.6 | 48.0 | 48.8 | 48.1 | 47.5 |
| | (14.8) | (16.7) | (15.3) | (15.4) | (15.7) | (15.6) | (16.9) | (16.9) | (14.6) | (17.4) | (14) | (16) |
| BMI, kg/m² (SD) | 31.9 | 27.4 | 30.8 | 32.2 | 33.6 | 31.3 | 31.1 | 28.2 | 32.5 | 28.6 | 34.0 | 30.8 |
| | (7.5) | (7.) | (8.7) | (7.5) | (9.9) | (6.9) | (7.4) | (6.2) | (8.6) | (6.1) | (8.6) | (7.8) |
| Sex (%) | | | | | | | | | | | | |
| Female | 133 (55.6) | | | 118 (77.6) | | 187 (71.6) | | 803 (69.7) | | 427 (55.5) | | 1668 (64.8) |
| Male | 106 (44.4) | | | 34 (22.4) | | 74 (28.4) | | 349 (30.3) | | 342 (44.5) | | 905 (35.2) |
| Smoking (%) | | | | | | | | | | | | |
| Ever | 86 (36) | | | 53 (34.8) | | 95 (36.4) | | 392 (34) | | 290 (37.7) | | 916 (35.6) |
| Never | 153 (64) | | | 99 (65.2) | | 166 (63.6) | | 760 (66) | | 479 (62.2) | | 1657 (64.4) |
| Ancestry (%) | | | | | | | | | | | | |
| AFR | 68 (28.5) | | | 37 (24.3) | | 28 (10.7) | | 966 (85.4) | | 206 (26.8) | | 1305 (50.7) |
| AMR | 107 (44.7) | | | 106 (69.7) | | 227 (87) | | 74 (6.4) | | 197 (25.6) | | 711 (27.6) |
| EUR | 41 (17.2) | | | - | | - | | 63 (5.5) | | 211 (27.4) | | 324 (12.6) |
| Other | 23 (9.6) | | | - | | - | | 49 (4.3) | | 155 (20.2) | | 233 (9.1) |

Metrics in table are shown as mean (SD) or count (%). Participants' ancestry was based on ancestry prediction information provided by All of Us. To protect participant privacy and in accordance with All of Us data and statistics dissemination policy, ancestry groups with fewer than 20 individuals are not shown. Empty categories may include zero or low-count values. For reporting purposes, we combined counts of sex, smoking status and ancestry across centers that used the same sequencing platform and coverage threshold. Centers marked with an asterisk (*) under the mid-pass (12x or 8x) column represent samples sequenced at a minimum of 8× coverage, whereas unmarked centers represent samples sequenced at a minimum of 12× coverage.

*BCM* Baylor College of Medicine, *BI* Broad Institute, *JHU* Johns Hopkins University, *UW* University of Washington, *HA* HudsonAlpha Institute, *AFR* African, *AMR* Latino/Admixed American, *EUR*: European.

## Variability in csTL across chromosome arms

Comparison of csTLs across all chromosome arms revealed significant variability attributable to arm (ANOVA: $F(38, 32,065) = 89.9$, $P < 2.2 \times 10^{-16}$), with some arms exhibiting longer TLs (3p, 4q, and 13q) and others tending to be shorter (3q, 16q, and 20q) across participants (Fig. 4a). Across PacBio-derived batches, we observed clear consistency in the ranking of chromosome arms by mean csTL (Fig. 4b).

We compared our ranking of arms by csTL to those reported by Karimian et al.[43], who used an ONT-based approach involving telomeric enrichment in peripheral blood mononuclear cells (PBMC) DNA samples from 147 individuals. They calculated relative csTL—defined as the csTL estimate normalized by each individual's mean TL across arms —and ranked arms by the relative csTL. Despite differences in methodology, we observed a strong correlation between our rankings and theirs (Spearman $\rho = 0.9$, $P < 2.2 \times 10^{-16}$). Several of the same chromosome arms appeared at the extremes in both studies, with 3p, 4q, and 12q being among the longest, and 17p, 12p, and 20q being among the shortest (Fig. 4c).

## Determinants of variation in csTL

We used linear mixed models (LMMs) to characterize sources of variability in csTL, adjusting for fixed effects of age, sex, ancestry, batch, BMI, and smoking status, and random effects of individual and chromosome arm. Our analysis revealed that csTLs vary substantially by individual, with 8.9% (chi-squared ($\chi^2$) likelihood ratio test (LRT): $\chi^2$ (1) = 1845.4, $P < 2.2 \times 10^{-16}$) of the variance in csTL attributable to inter-individual differences. Individuals with longer mean TL tend to have systematically longer csTL across all chromosome arms (Fig. 5a and Supplementary Data 1). Chromosome arm accounted for 9.1% (LRT: $\chi^2$ (1) = 3751.7, $P < 2.2 \times 10^{-16}$) of the variance in TL, highlighting clear differences in TL across chromosomes (Figs. 4a and 5b, Supplementary Data 1).

To assess the contribution of each fixed effect, we fit LMMs adjusting for the random effects of individual and chromosome arm and introduced each fixed effect separately. Among the fixed effects, batch explained the most variance in csTL (14.6%, LRT: $\chi^2$ (8) = 1359.2, $P = 4 \times 10^{-288}$), followed by age (3.7%, LRT: $\chi^2$ (1) = 268.5, $P = 2 \times 10^{-60}$), ancestry (3.1%, LRT: $\chi^2$ (5) = 228.3, $P = 2 \times 10^{-47}$), sex (1.1%, LRT: $\chi^2$ (1) = 74.4, $P = 6 \times 10^{-18}$), smoking status (0.4%, LRT: $\chi^2$ (1) = 23.9, $P = 1 \times 10^{-6}$) and BMI (0.2%, LRT: $\chi^2$ (1) = 15.6, $P = 7 \times 10^{-5}$) (Fig. 5b and Supplementary Data 1). The combined contribution of all fixed effects to variation in TL was 18.7% (LRT: $\chi^2$ (17) = 1886.9, $P < 2.2 \times 10^{-16}$).

To conduct analyses focused solely on biological sources of variation, we modeled csTL as a function of batch and used the residuals as a csTL estimate with batch-related variation removed. In analyses of batch-adjusted csTL residuals, the contribution of ancestry was substantially attenuated (0.1%, LRT: $\chi^2$ (5) = 10.3, $P = 0.07$). Batch was strongly correlated with ancestry—with some batches comprised entirely of participants from a single ancestry—and BMI; therefore, even with careful batch adjustment, we were likely limited in our ability to accurately assess the contribution of ancestry and BMI to variation in csTL due to confounding by batch (Fig. 5c and Supplementary Data 1–3). In this analysis, the contributions of individual (10.6%, LRT: $\chi^2$ (1) = 1906.4, $P < 2.2 \times 10^{-16}$) and chromosome arm (10.7%, LRT: $\chi^2$ (1) = 3745.1, $P < 2.2 \times 10^{-16}$) increased, and the contributions of age (3.9%, LRT: $\chi^2$ (1) = 445.9, $P = 6 \times 10^{-99}$), sex (0.7%, LRT: $\chi^2$ (1) = 68.2, $P = 1 \times 10^{-16}$), and smoking (0.2%, $\chi^2$ (1) = 25.8, $P = 4 \times 10^{-7}$) persisted (Fig. 5c and Supplementary Data 2). The combined contribution of all fixed effects to variation in TL after pre-adjusting for batch was 4.4% (LRT: $\chi^2$ (9) = 513.8, $P < 2.2 \times 10^{-16}$).

In chromosome-specific analyses, ancestry showed weak associations with csTL when comparing AFR (reference) to AMR and EUR ancestry groups (Supplementary Data 4). While most arms did not show associations ($P > 0.05$), binomial tests showed a consistent pattern in the direction of association, with shorter csTLs in AMR ($\hat{p} = 0.79$, 95% confidence interval (CI) = [0.66, 1], $P = 0.0001$) and EUR ($\hat{p} = 0.74$, CI = [0.6, 1], $P = 0.002$) individuals compared to AFR. Sex and smoking also exhibited weak associations with csTL at individual arms (Supplementary Data 5 and 6, respectively). However, binomial tests indicated that males tend to have shorter csTLs than females ($\hat{p} = 1$,

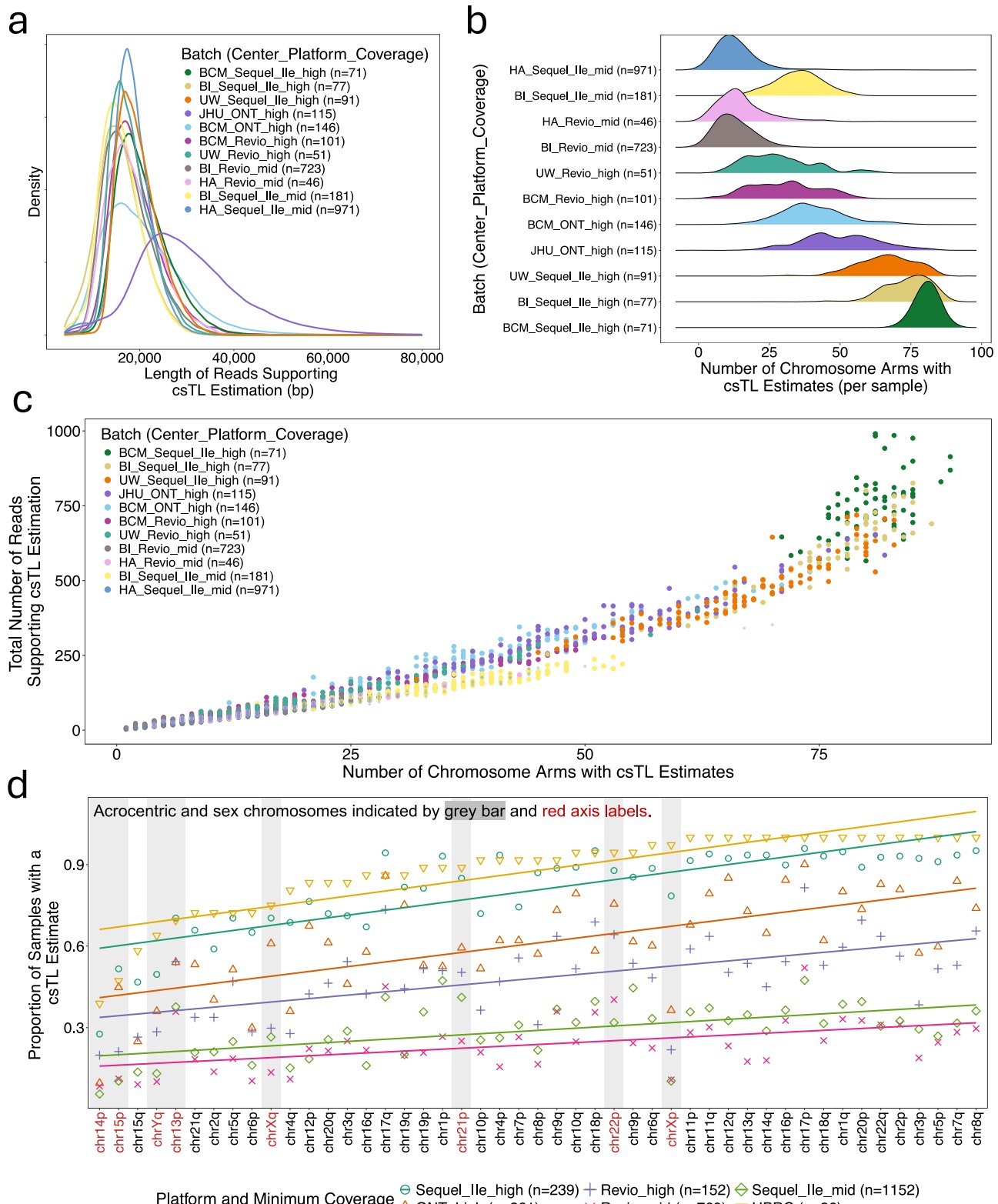

**Fig. 2 | Telogator2 performance varies by sequencing coverage, platform, and chromosome arm. a** Density plot showing the distribution of supporting read lengths (reads mapped to telomeric regions) across batches. Read lengths >80 kb not shown. **b** Ridge plot showing the distribution of the number of chromosome arms with csTL estimates per sample across batches. **c** Scatterplot of telomere coverage (number of reads supporting csTL estimation) versus the number of chromosome arms with csTL estimates. Each dot is an individual and colors denote batch. **d** Scatterplot showing the proportion of samples with a csTL estimate for each chromosome arm (ordered by proportions observed in the Human Pangenome Reference Consortium dataset). Dot color and shape denote data source (HPRC or All of Us aggregated batches defined by sequencing platform and coverage threshold). chrYp is not displayed because estimates were available for only five All of Us samples and none of the HPRC samples.

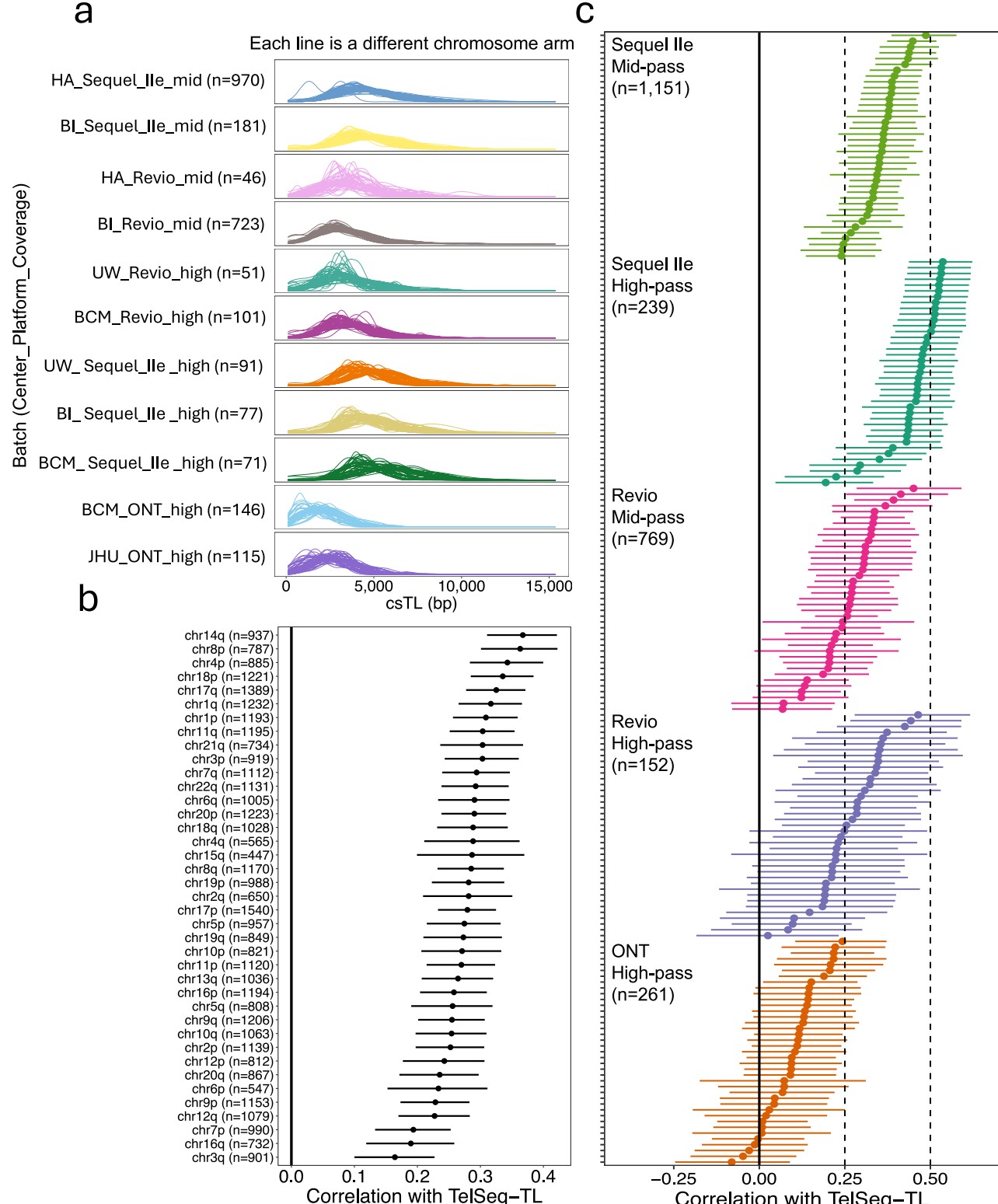

**Fig. 3 | csTL distribution and correlation with adjusted TelSeq-TL varies by sequencing platform and coverage. a** Density plot showing the distribution of csTLs across batches. Each line is a unique chromosome arm (**b**) Forest plot showing Pearson correlation coefficients (point) and 95% confidence intervals (horizontal bar) between principal component (PC) adjusted TelSeq-TL and csTL for all participants (n = 2573). **c** Forest plot showing Pearson correlation coefficients (point) and 95% confidence intervals (horizontal bar) between PC-adjusted TelSeq-TL and csTL calculated by batch. Batches are defined by sequencing platform and minimum coverage threshold. Acrocentric and sex chromosomes are not displayed.

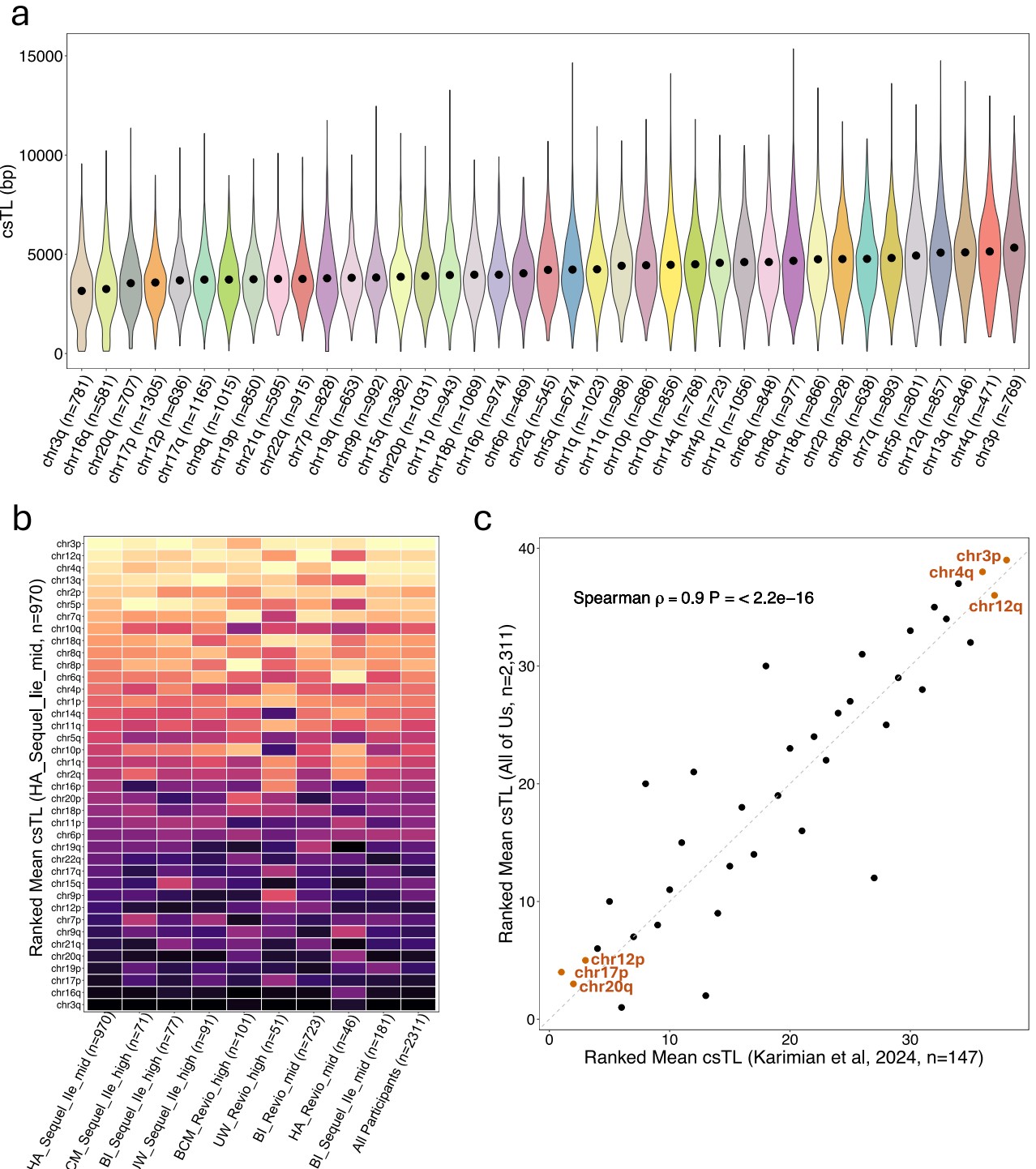

**Fig. 4 | Mean csTLs differ across chromosome arms. a** Violin plot displaying the distribution of csTL for each chromosome arm across all individuals (excluding ONT batches; *n* = 2311), ordered by the mean csTL (denoted by a black dot). **b** Heatmap comparing the rank of chromosome arms based on mean csTLs across PacBio (Sequel2e and Revio) batches and all participants. Chromosomes are ordered from top to bottom in decreasing order of mean csTL as determined from HA_Sequel2e_mid batch (*n* = 970). Color intensity represents the rank of each chromosome arm within each batch, from short (dark) to long (light). **c** Scatterplot comparing the rank order of chromosome arms based on mean csTLs across All of Us participants (*n* = 2311) with the rank order of arms based on mean relative csTL from Karimian et al.[43] (*n* = 147). Spearmen's rank correlation coefficient (*ρ*) and *P*-value from the two-sided correlation test are displayed.

CI = [0.92, 1], $P = 2 \times 10^{-12}$) and that ever smokers tend to have shorter csTLs compared to never smokers ($\hat{p} = 0.67$, CI = [0.52, 1], $P = 0.03$).

For all chromosome arms, we observed an inverse association between age and csTL (Fig. 6a). These associations remained significant after including adjustments for BMI, sex, ancestry, smoking, and batch (Supplementary Data 7). We identified a significant interaction between

age and chromosome arm (LRT: $\chi^2$ (2) = 72.7, $P = 2 \times 10^{-16}$), indicating that the association of age with csTL varies by arm (Fig. 6b and Supplementary Data 8). To confirm this heterogeneity in age-csTL associations was consistent across batches, we evaluated age-csTL associations for the Revio and Sequel IIe batches separately and observed a positive correlation between the Revio and Sequel IIe beta coefficients (Pearson

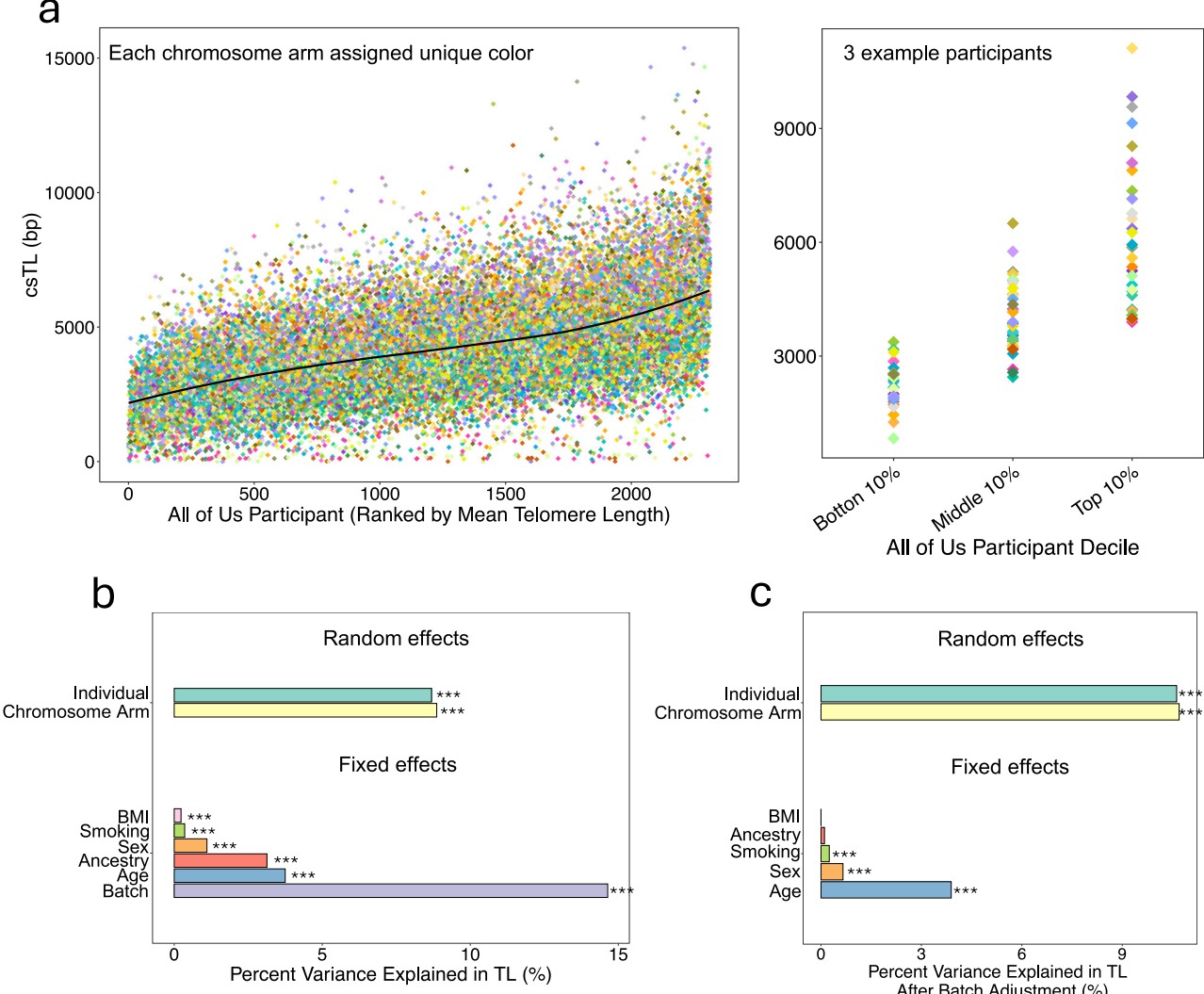

**Fig. 5 | csTLs vary across participants and by participant characteristics. a** Left: Distribution of csTL across all participants (*n* = 2311), ranked by participant's mean TL across all measured chromosome arms. Right: csTL profiles for 3 participants selected based on their ranking by individual mean TL: participant ranked 63 (bottom 10%), 1120 (middle 10%), and 2263 (top 10%) displayed. Each chromosome arm is assigned a unique color. **b** Sources of variation in TL and their relative contributions, based on a linear mixed model including fixed and random effects. **c** Sources of variation in residual TL and relative contributions, based on a linear mixed model including fixed and random effects. Residual TL is the variation in TL after batch adjustment. Chi-squared likelihood ratio test *P* values * < 0.01, **<0.001, ***<0.0001 are displayed. Exact *P*-values for fixed and random effects in (**b**, **c**) are provided in Supplementary Data 1 and 2, respectively.

*r* = 0.4, *P* = 0.01) (Fig. 6c). Interestingly, the magnitudes of the age-related beta coefficients were positively correlated with arm-specific mean csTLs (Pearson *r* = 0.8, *P* = 4 × 10⁻⁹), such that arms with longer csTL exhibited a stronger association with age (Fig. 6d).

**Associations between chronic disease status and csTL**
Using electronic health record (EHR) data from the AoU study, we examined the association between chronic disease status and csTL. We focused on cardiovascular disease (CVD)—cases defined as individuals with hypertension, ischemic heart disease, or heart failure—and type 2 diabetes (T2D) due to their established links to TL and sufficient case numbers for analysis.

We did not observe significant associations between csTL and either CVD or T2D status across chromosome arms or for any individual arms (*P* > 0.05) (Supplementary Fig. 7). This lack of association could be due to limited power at the chromosome-arm level or because much of disease-related variability was already captured by inter-individual differences modeled as a random effect.

To explore additional biologically relevant TL metrics, we analyzed disease associations using (1) the mean TL, a commonly studied measure, and (2) the shortest TL per individual, which has been hypothesized to better reflect cell function and viability, but has rarely been examined in population studies. Although the associations did not reach statistical significance, our results suggested longer TL may be associated with reduced disease risk (Supplementary Data 9). For example, each 1 kb increase in shortest TL (odds ratio (OR) = 0.94, CI = [0.84, 1.05], *P* = 0.26) and mean TL (OR = 0.94, CI = [0.82, 1.08], *P* = 0.4) corresponded to lower odds of CVD risk. Similar patterns were observed for hypertension, which accounted for the majority of CVD cases (shortest TL: OR = 0.95, CI = [0.85, 1.06], *P* = 0.34; mean TL: OR = 0.96, CI = [0.83, 1.10], *P* = 0.56). For type 2 diabetes (T2D), associations were weaker (shortest TL: OR = 1.03, CI = [0.91, 1.17], *P* = 0.66; mean TL: OR = 1.00, CI = [0.85, 1.18], *P* = 0.95). These preliminary results suggest that the shortest TL may offer additional insights for disease susceptibility, warranting further exploration in larger cohorts.

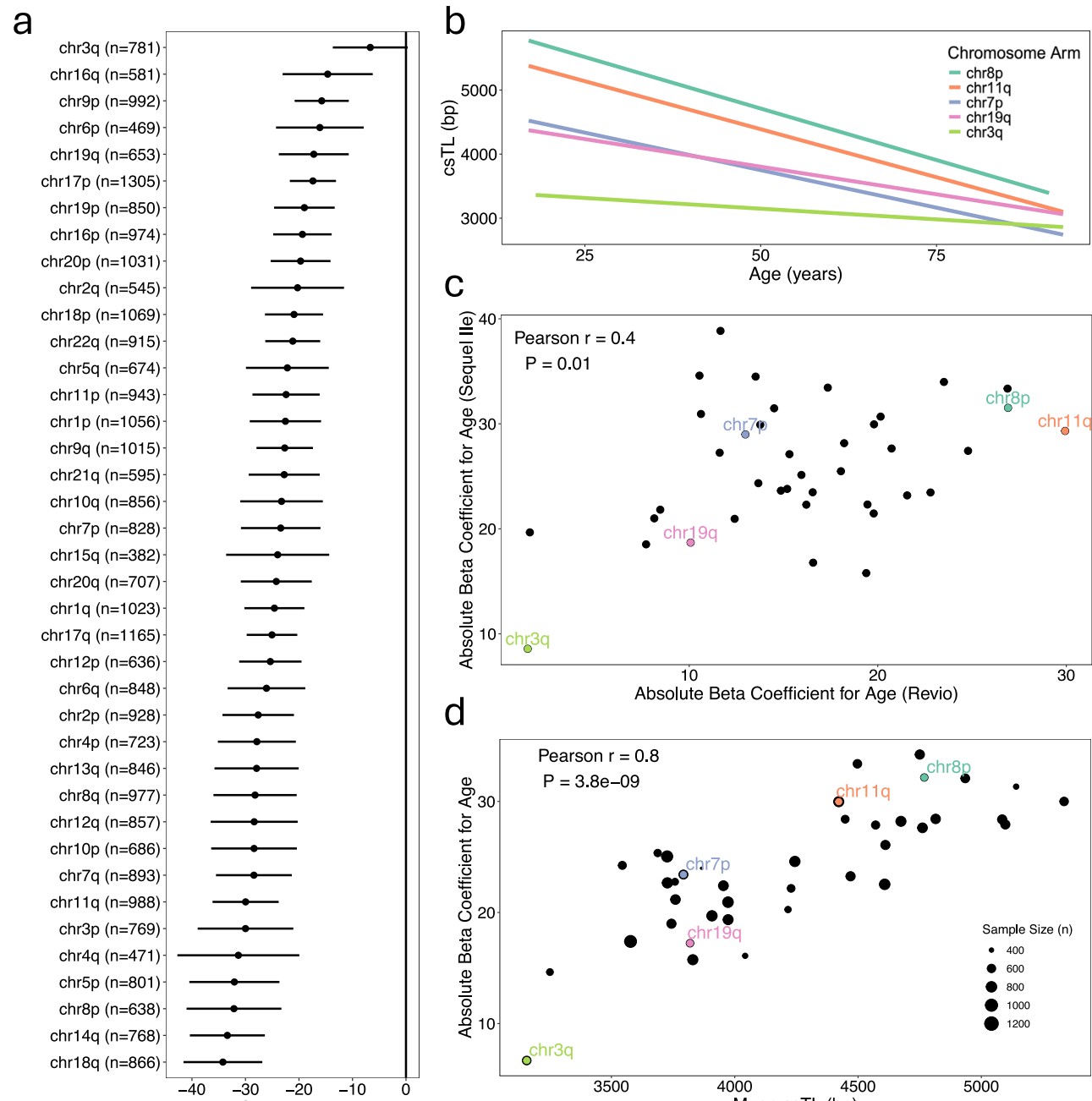

**Fig. 6 | Age is negatively associated with telomere length across all chromosome arms. a** Forest plot showing estimated beta coefficients (point) and 95% confidence intervals (horizontal bar) from linear models examining the association between age (years) and csTL for each chromosome arm, across all participants (*n* = 2311). **b** Association between csTL and age for five representative chromosome arms. **c** Scatterplot comparing the estimated absolute beta coefficients for age from linear models fitted separately to Sequel IIe and Revio samples. Pearson correlation coefficient (*r*) and *P*-value from two-sided correlation test displayed. **d** Scatterplot comparing the mean csTL for each chromosome arm and estimated absolute beta coefficients for age for each arm. Pearson correlation coefficient (*r*) and P-value from two-sided correlation test displayed.

## Discussion

In this study, we estimated csTLs for 2573 participants from the AoU cohort by applying Telogator2 to lrWGS data. We evaluated the impact of lrWGS metrics on Telogator2's performance and characterized sequencing coverage and platform as key factors influencing csTL estimation. Additionally, we showed that csTL varies substantially by chromosome arm, with our ranking of arms by mean csTL closely mirroring rankings reported in prior studies. We demonstrated that variation in csTL is largely attributable to chromosome arm and individual, with the majority of the individual effect being independent of

the age effect. Age remains a clear contributor and is negatively associated with csTL across all arms; however, the magnitude of the association varies by arm, with longer arms showing stronger associations with age. Lastly, we demonstrated the utility of estimating csTL for disease association studies, enabling examination of the shortest telomere as a risk factor.

We observed substantial variation in csTL for any given chromosome arm, with csTL ranging from ~0.1 to ~20 kb (Fig. 4a). This variability aligns with findings from previous studies employing DNA-microarray FISH[55] and lrWGS methodologies[43,44] to estimate csTL.

Notably, the detection of csTLs as short as a few hundred bp raises important questions about the minimal TL required for maintaining cellular viability. For instance, a recent study in yeast demonstrated that telomeres as short as ~75 bp may still support cellular function under certain conditions[56]. However, it's important to acknowledge that Telogator2 may underestimate true csTLs due to technical limitations, such as reads aligning to the subtelomere but not extending to the end of the telomere or DNA degradation during library preparation. These limitations are especially relevant when csTL estimates are at the sub-kb level. Thus, while the detection of extremely short telomeres is potentially informative, the reliability of such measurements remains subject to technical uncertainty and should be interpreted with caution. Additionally, because most individuals have a single csTL estimate for a given arm—likely driven by the large subset of low telomeric coverage samples in AoU (Fig. 2c)—our ability to assess haplotype-specific differences in csTLs was limited, highlighting the need for future studies with improved telomeric coverage to allow for this type of analysis.

Despite sufficient coverage and read lengths, ONT batches were less reliable for csTL estimation, producing systematically shorter csTL estimates and weaker correlations with TelSeq-TL (Fig. 3c) and age (Supplementary Fig. 8) compared to PacBio batches (Revio and Sequel IIe). This reduced reliability could be due to ONT's higher base-calling error rates in repetitive regions. For this reason, the Telogator2 developers noted limitations when analyzing telomeres using older base callers like Guppy[42,57]. Consistently, a recent study showed that ONT reads exhibit systematic miscalls within telomeric repeat regions (e.g., TTAGGG), a phenomenon not observed in PacBio sequencing reads[58]. It is therefore plausible that ONT-specific biases lead to inaccurate identification of telomere−TVR boundaries such that miscalled telomeric repeats are not recognized as part of telomeric regions, resulting in underestimation of csTLs. The AoU ONT data include reads processed with both Guppy and the newer Dorado base caller, which provides improved accuracy; however, the available metadata did not allow us to distinguish between them. Additionally, because raw reads were unavailable, we could not re-call bases using newer or telomere-trained models[57]. Beyond base-calling errors, DNA extraction and library preparation protocols could also influence estimation, though future studies are needed to clarify their effects. Together, these findings highlight ongoing technical challenges in using AoU lrWGS data for csTL estimation.

The protein-counting model, a predominant model of TL regulation, proposes that telomeres are regulated around a similar mean distribution through preferential elongation of short telomeres and inhibitory effect of telomere-bound proteins on telomerase access to longer telomeres[59]. This TL equilibrium across arms is believed to be established at birth and maintained throughout life. Notably, the model appears consistent with our observation that the magnitude of the age-csTL association varies according to arm length, potentially reflecting preferential maintenance of shorter telomeres (discussed in more detail below).

However, our analysis also revealed significant differences in TL across arms and a stable ranking of chromosome arms based on their mean TL across individuals. These patterns mirror those reported by Karimian et al.[43], despite differences in population and cell composition—AoU used whole blood, whereas Karimian et al.[43] focused on PBMCs. Thus, while the protein-counting model provides a general framework for telomere homeostasis, the persistent differences in TL across arms points to additional regulatory mechanisms contributing to csTL variability.

One potential contributor to differences in TL across arms is the subtelomere region, which differs in sequence between arms. Recent work suggests that subtelomeric chromatin can influence telomerase recruitment in a cis-acting, telomere-specific manner[60]. Epigenetic modifications at subtelomeres may also contribute to TL regulation.

Changes in DNA methylation patterns at the subtelomere, a heavily methylated region, have been linked to changes in TL regulation[61]. For instance, a prior study showed that subtelomeric hypomethylation in DNA methyltransferase-deficient mouse embryonic cells resulted in elongated telomeres[62]. These findings highlight the importance of additional research analyzing variability in csTL regulation across arms to better understand the diversity of TL regulatory mechanisms. Future research should also investigate the consistency of csTL patterns in other populations and tissue/cell types to better understand how genetic, epigenetic, and cellular context shape csTL variation.

We found that a substantial proportion of the variance in csTL (25.7%) was explained by biological factors. Specifically, a large proportion of the variance in csTL was attributable to individual, independent of the age effect, supporting the hypothesis that individuals are born with an inherently short or long telomere status across arms that is maintained across the lifespan[63].

Age is one of the most well-established determinants of TL and showed a clear inverse association with csTLs across all arms. The strength of this association varied by arm, suggesting that telomere attrition may not occur at a uniform rate across arms. Arm-specific age-csTL association estimates were strongly correlated with arm-specific mean csTL, suggesting that arms with longer csTL shorten more rapidly with age. This pattern is consistent with models in which telomere maintenance preferentially acts on shorter telomeres, thereby buffering them against age-related attrition and making age-related shortening more pronounced for longer telomeres. These dynamics may, in turn, allow cells to undergo more divisions before reaching a critically short TL that triggers cell senescence. Chromosome-specific replication and chromatin features[64] could also contribute to this observed pattern. Although our read lengths (~20 kb for PacBio batches) should capture most telomeres, very long telomeres—likely more common in younger individuals—may be underestimated, which could lead to some attenuation of the age-related associations. Longitudinal studies will be needed to validate our findings and better characterize csTL dynamics over time.

We assessed the associations of additional established determinants of TL, including ancestry, sex, BMI, and smoking status, confirming their significant contributions to TL variation across arms. While we did not observe statistically significant associations for these factors at individual arms, we did observe associations that were consistent with multiple prior studies[13,65−67] when analyzing all arms. Specifically, individuals of AFR ancestry tended to have longer telomeres across all arms compared to individuals of AMR and EUR ancestry. Similarly, never-smokers and females generally exhibited longer TL across arms. Our inability to detect associations for individual arms was likely related to limited sample sizes. Nevertheless, the consistency we observe with established associations highlights the reliability of csTL estimates obtained from lrWGS data for population-based telomere research.

csTL profiling allowed us to examine associations between TL and EHR-based chronic disease phenotypes at individual chromosome arms, as well as with mean TL (averaged across all measured arms per individual) and the shortest TL per individual. We focused our analysis on CVD and T2D, given their known associations with TL and sufficient case numbers within our AoU sub-cohort. We did not detect significant associations for either disease at the individual arm-level, nor with mean TL or shortest TL, likely due to power limitations. However, in line with previous studies[68,69], we observed suggestive trends indicating longer TL was associated with lower odds of CVD and hypertension risk. This trend was present when considering the shortest TL per individual, a metric not commonly evaluated, highlighting its potential utility as a relevant biomarker for disease risk.

Findings from studies in yeast[20−22] and mice[20], have shown that the length of the shortest telomere is a critical determinant of replicative senescence, cellular viability, and chromosomal stability. Small studies

of humans also suggest that the shortest telomere (or length at individual telomeres) may be relevant biomarkers for disease. For instance, a prior study of 13 individuals with CVD reported shorter lengths of the median and short telomere (20th percentile TL) in patients compared to age-matched healthy controls[24]. Another study reported enrichment of critically short telomeres amongst individuals (*n* = 52) with myelodysplastic syndrome (MDS), a clonal disorder of hematopoietic stem cells[26], while another reported that short TL at certain arms (e.g., chr17p and chr12q) but not others (e.g., chr11q and chr2p) is associated with esophageal carcinogenesis[70].

Beyond assessing the mean and shortest TL, csTL profiling allows for the exploration of additional metrics—such as the proportion and mean lengths of both short and long telomeres within individuals. While previous studies in large human cohorts have primarily relied on mean TL to report associations with disease risk (e.g., shorter mean TL with increased risk of CVD, and longer mean TL with certain types of cancer), such approaches cannot examine variation in TL across arms. Given the suggestive trends observed in our analysis, and evidence from prior studies, we propose that incorporating csTL estimates into large research studies of diverse human population cohorts, including assessment of TL distributions and extreme values (both short and long), has the potential to provide important insights into the variation of csTLs across individuals and the roles of csTLs in human aging and disease. At the same time, it is important to recognize that high-quality mean TL measurements from single patients using accurate, clinically-validated methods show clear utility for diagnostics and treatment decisions[71]. While mean TL will continue to be an important measure in improving our understanding of human aging and disease biology, complementing it with large-scale csTL analyses will further bridge research findings with clinical applications.

As the utilization of long-read sequencing technologies increases, lrWGS will likely become a common approach for measuring genetic variation in human cohorts. This study provides a useful guide to researchers interested in the expected performance of csTL estimation as applied to emerging lrWGS datasets. Despite the limitations of our study, such as sample size, platform heterogeneity, and low telomeric coverage for many samples, we were able to characterize biologically meaningful variation in TL across chromosome arms for >2500 individuals.

## Methods

This study used de-identified data from the All of Us Research Program. Informed consent was obtained by All of Us from all participants, and the All of Us Research Program protocol was reviewed by the NIH All of Us Institutional Review Board.

### AoU cohort

AoU is a longitudinal cohort study that aims to collect electronic health records, genomics, and survey data for one million US participants, including individuals historically underrepresented in biomedical research. A detailed description of the AoU cohort and research protocol has been previously published[72,73].

We utilized AoU data on blood-derived lrWGS and srWGS, demographics, and health-related traits. The AoU v8 release (February 2025) includes lrWGS data for 2842 samples, comprising 1815 newly added and 1027 carried over from v7 (April 2023). We excluded 60 samples flagged by AoU based on various quality control (QC) metrics[74]. Additionally, 42 individuals in v8 were sequenced twice, using two different sequencing platforms, resulting in 2740 unique individuals with lrWGS data.

Sequencing was performed using PacBio Revio, PacBio Sequel II/IIe, or ONT R10.4 on PromethION across five facilities: BCM, BI, JHU, UW, and HA. Participants were selected for lrWGS by each sequencing facility, with the main criterion being that each participant had matching srWGS data. The v8 release includes 414,830 srWGS

samples, comprising 171,436 newly added and 243,394 from v7. QC procedures for v7 and v8 releases of genomic data are detailed in the AoU QC reports[74,75].

### Estimation of csTLs using Telogator2

We created a Docker container for Telogator2 (niyatij/telogator2_image) to allow integration with the workflow management system, Cromwell[76], on the AoU workbench. This enabled more efficient, parallel processing of the lrWGS data. Briefly, Telogator2 extracts telomere reads that contain the canonical telomeric repeats TTAGGG, clusters them based on telomere variant repeats (region between telomere and subtelomere, comprised of canonical repeats interspersed with blocks of variant repeats) to identify individual telomere clusters, and aligns them to T2T reference genomes at subtelomere boundaries to estimate TL from the boundary to the read end for each chromosome arm (*n* = 92 for diploid human genome)[42].

To optimize analysis for PacBio data, we compared two input formats: BAM and hifiasm assembly files. Hifiasm[77], a de novo assembly software, generates fully phased haplotype assemblies, effectively producing a single representative read per telomere, which we hypothesized could improve processing efficiency due to its smaller file size. However, despite this advantage, hifiasm files did not perform as well as BAM files, which exhibited stronger correlations with age (known correlate of TL) (Supplementary Fig. 9) and TelSeq-TL estimates (Supplementary Fig. 10). Additionally, given the variation in sequencing coverage and platforms across our samples, we adjusted the parameter minimum number of reads for cluster accordingly: -n 4 for high-coverage samples (>25×), -n 3 for low-coverage samples (< 20×), and -n 1 for hifiasm files, specifying both HiFi and ONT sequencing technologies. All downstream analyses used the default TL estimate (75th percentile TL).

Of the 42 duplicate samples, 41 were sequenced using both PacBio (either Revio or Sequel IIe) and ONT, and one was sequenced using both PacBio Revio and Sequel IIe. For the 41 samples, we present the PacBio data as they provided more csTL estimates. For the single sample, we present the Sequel IIe data.

Unless otherwise specified, we present the average csTL for chromosome arms with multiple csTL estimates. In the human diploid genome, two csTL estimates are expected per chromosome arm. In the majority of individuals, for any given chromosome arm with an estimate, there were no more than two csTL estimates (Supplementary Fig. 4). Therefore, when two (or more) estimates were available for a given arm, we averaged them; when only one estimate was available, the single estimate was used as csTL. After averaging, each participant had up to 48 csTL estimates (22 autosomal pairs × 2 + 4 sex chromosomes), representing the diploid genome's 92 chromosome arms (22 autosomal pairs × 4 + 4 sex chromosome arms). We excluded csTL estimates shorter than 100 base pairs (bp), TVR (telomere variant repeat) lengths less than 0, and csTL estimates not clearly mapping to a single chromosome arm (i.e., ambiguous mapping to multiple arms).

### Estimation of average TL using TelSeq

We used a publicly available Docker container (jweinstk/telseq) to facilitate processing of the srWGS data to generate average TL estimates. The container runs a pipeline that takes in a list of CRAM files, converts them to BAM format using SAMtools[78], and runs TelSeq[32] for average TL estimation. It also integrates with workflow management system Cromwell[76] for parallel processing. The TelSeq parameter k was set to 12 (read length 151 bp) in line with previous studies[6,8,54]. We ran this pipeline for individuals with lrWGS data (*n* = 2740).

Additionally, to account for technical sources of variability, we adjusted our TelSeq-TL with sequencing depth data, as described previously[6,8,54]. Briefly, we ran Mosdepth[79] on srWGS CRAM file in bins of 1000 bp across the genome. Next, we generated PCs from the sequencing depth data using NGS-PCA, excluding regions with low

mappability and a curated list of known problematic structural variants. We calculated batch-adjusted estimates by regressing out the top 30 batch PCs and used the residuals as the adjusted TelSeq-TL. To validate the reliability of our TelSeq-TL estimates, we confirmed that both the unadjusted TelSeq-TL ($R = -0.4$, $P < 2 \times 10^{-16}$) and adjusted TelSeq-TL ($R = -0.53$, $P < 2 \times 10^{-16}$) showed the expected negative correlation with age, with a stronger correlation after batch adjustment (Supplementary Fig. 11A, B, respectively).

## AoU covariates

Age, sex, smoking status, batch, and genetic ancestry (EUR, AFR AMR, EAS, SAS, and MID) were used as covariates in our regression models. Age, sex, and smoking status were extracted from concepts sets on the AoU workbench. Age (years) was calculated using the lubridate package[80] in R (R/4.4.0), by subtracting the participant's birth date (Demographics concept set) from the date of blood sample collection for genomic sequencing (auxiliary genomics metric file). Sex was extracted from the sex_at_birth field under the Demographics concept set. This variable reflects self-reported sex collected at enrollment in the AoU Research Program. Body mass index (BMI) was calculated based on individual height and weight (kg/m²). Smoking status was classified as ever for participants who reported yes to smoking 100 or more cigarettes in their lifetime, encompassing both current and former smokers, while never smokers were those who reported no to smoking fewer than 100 cigarettes in their lifetime. Ancestry information was derived from the ancestry prediction file provided by AoU. Ancestry predictions were generated using a PCA-based approach that incorporates a random forest classifier trained on Human Genome Diversity Project (HGDP) and 1000 genomes variants from Genome Aggregation Database (gnomAD)[72].

Batches were defined according to the batch classifications specified in the AoU v8 QC report[74]. Batches were organized based on sequencing platform (PacBio Sequel IIe, PacBio Revio, and ONT), sequencing center (HA, BCM, JHU, UW, and BI), data release (v7 and v8), and minimum coverage (8x for v7, 12x for v8, both classified as mid-pass, and high-pass 25x). For some downstream analyses and visualizations, results were aggregated across centers, yielding five aggregated batches defined based on the sequencing platform and minimum coverage threshold.

For all analyses, individuals with missing data on any of the following covariates—ancestry prediction, age, sex, BMI, or smoking status—were excluded, resulting in a sample size of 2573 individuals.

## Long-read WGS metrics and Telogator2 performance

Metrics related to sequencing coverage of lrWGS data were extracted from the auxiliary genomic metric files provided by AoU. TL-related metrics were derived from the Telogator2 output file, and the TL_p75 and read_lengths metrics were used to evaluate Telogator2 performance across different sequencing coverage levels, sequencing platforms, and chromosome arms. To further evaluate Telogator2 performance, we analyzed 36 high-coverage samples from (>30× telomere coverage) the HPRC (Supplementary Data 10), that were sequenced using the PacBio Revio platform[50]. Initial coverage of the samples was calculated using Mosdepth[79]. To facilitate comparison with AoU csTL estimates, we then systematically down-sampled the BAM read data to 5×, 10×, 15×, 20×, and 25× telomeric coverage by determining the fraction of reads needed to achieve the target coverage. Down-sampling was performed using the software Sambamba[81], and we assessed the correlation (Pearson correlation) and absolute mean error in Telogator2 csTL estimates between high coverage and down-sampled data, considering only csTL estimates that were present in both. Additionally, the HPRC dataset served as an independent reference to evaluate whether differences in Telogator2 performance across chromosome arms were consistent across datasets.

## Estimation of variance in csTL explained by covariates

We used LMMs to estimate the contribution of covariates to the variance in csTL. Our approach was based on methods presented by Demanelis et al.[3], which followed the framework proposed by Nakagawa et al.[82] and utilized the multi-model inference (MuMIn) package and lme4 package in R (R/4.4.0).

We fit an LMM to estimate the percent variance explained (PVE) in csTL by fixed-effects (age, sex, ancestry, BMI, batch, and smoking status) and random effects (chromosome arm and participant). The variance explained by all fixed effects was calculated as the variance of the fitted values, obtained by multiplying the design matrix of fixed-effect covariates by the vector of fixed-effect coefficients. The variance for each of the random effects (chromosome arm and participant) was extracted from the model output using lme4::VarCorr. The PVE for each component was then computed by dividing its variance by the total variance, which included the variance of the combined fixed effects, the variance of each random effect, and the residual variance. To test the significance of each component, we used LRTs comparing full models to reduced models excluding all fixed effects, random effects of chromosome arm, or participant, respectively.

To assess the contribution of individual fixed-effects (representing age, sex, BMI, ancestry, smoking, and batch) to the variation in TL, we compared two LMMs: one that included a single fixed effect and the random effects of chromosome arm and participant, and another that included only the random effects without the single fixed effect. The contribution to variance explained in TL was derived from the marginal $R^2$ (which captures variance explained by the fixed effect). We extracted the $P$-value from an LRT comparing these two models.

Given that batch is correlated with ancestry and BMI, we compared three batch-correction approaches, modeling batch (i) as a fixed effect, (ii) as a random effect, and (iii) by pre-adjusting TL for batch prior to analysis. The statistical significance ($P$-values), the direction and magnitude of effect sizes for each covariate were generally consistent across the three approaches. We present results from models with batch included as a fixed-effect (allowing it to compete with ancestry and BMI) and where TL is modeled as function of batch, yielding TL residuals (isolating variation explained by biologic sources) in Supplementary Data 1 and 2, respectively. Results from batch modeled as a random effect is included as a sensitivity analysis (Supplementary Data 3).

## Chromosome arm-specific analyses of TL

Pearson correlations and linear regression were used to examine relationships between TL at individual chromosome arms and fixed-effect covariates (age, sex, smoking, BMI, and ancestry). Similar to the overall analysis, we compared three batch-correction approaches. We present results from analyses with batch modeled as (1) a fixed-effect, (2) where TL is modeled as a function of batch and (3) a random effect (Supplementary Data 4–7). We used a binomial test to assess whether the association between fixed-effect covariate and TL was predominantly in one direction (positive or negative) across chromosome arms more often than expected by chance. Additionally, to assess whether the association of fixed-effect covariates on TL varied by arm, we compared two LMMs: one including all fixed effects covariates and the random effects of chromosome arm and participant, and another incorporating an interaction term between the single fixed-effect covariate and random effect of chromosome arm, allowing the fixed effect to have different slopes and intercepts for each chromosome arm. We extracted the p-value from an LRT comparing these two models.

## Associations between chronic disease status and TL

Based on a prior AoU study, we extracted ICD-9 and ICD-10 codes for hypertension, ischemic heart disease, and heart failure[83] and defined a

CVD case as an individual with hypertension, ischemic heart disease, or heart failure. T2D status was similarly retrieved using concept ID codes from EHR. A full list of codes can be found in Supplementary Table 1. We examined the relationship between csTL and disease status across chromosome arms using a LMM, adjusted for fixed effects of age, sex, BMI, smoking, ancestry, batch and random effects of individual and chromosome arm. For each arm, we assessed associations between TL and disease status, adjusting for age, sex, BMI, smoking, ancestry and batch. We also evaluated the association of disease status and both mean TL (each individual's average TL across all measured arms) and the shortest TL (the minimum TL per individual). Individuals with mean or shortest TL values more than 3 standard deviations from the cohort mean were excluded. We used logistic regression, adjusted for age, sex, BMI, smoking, ancestry, and batch, to model disease status as a function of mean TL or shortest TL, estimating odds ratios and P values per 1-kb increase in TL.

### Reporting summary

Further information on research design is available in the Nature Portfolio Reporting Summary linked to this article.

## Data availability

The data generated in this study, including the chromosome-specific TLs from Telogator2 and TelSeq TLs, have been deposited in the community workspace titled Jain_et_al_csTL_Nature_Comm, accessed within the featured workspace collection of the All of Us Researcher Workbench (https://support.researchallofus.org/hc/en-us/articles/360059633052-Featured-Workspaces). Access to this data is available to researchers affiliated with institutions that have signed a Data Use agreement with the All of Us Research program and who have obtained controlled tier access (https://www.researchallofus.org/register/). Supplementary Data 10 includes the list of Human Pangenome Reference Consortium samples analyzed, along with links for accessing the data.

## Code availability

The code used to generate chromosome-specific telomere length estimates with Telogator2 is available on GitHub: https://github.com/niyati1211/All-of-Us-chromosome-specific-telomere-lengths[84]. The code used to produce the main Figures and Supplementary Figs. have been deposited in the community workspace titled Jain_et_al_csTL_Nature_Comm, accessed within the featured workspace collection of the All of Us Researcher Workbench.

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

## Acknowledgements

We gratefully acknowledge All of Us participants, who provided data and biological samples that supported this research. We also thank the National Institutes of Health's All of Us Research Program for enabling access to the data examined in this study. This work was supported by grants U01HG007601 (to B.L.P.), R35ES028379 (to B.L.P.), 1R01GM154421 (to L.S.C.), 1U01MH139345 (to L.S.C.), P30ES027792 (to H.A.), R03HL172114 (to B.A.), and 1OT2OD036445 (to B.A. and H.A.).

## Author contributions

N.J. contributed to study conception, performed analyses, interpreted the data, and wrote the main manuscript text.

## Competing interests

The authors declare no competing interests
