## [Transparent Peer Review file · Nature Communications]

Determinants of chromosome-specific telomere lengths among 2,573 All of Us participants

Corresponding Author: Ms Niyati Jain

Version 0:

Reviewer comments:

Reviewer #1

(Remarks to the Author)

I find this work to be interesting and an exciting application of the AoU genomic data as it leverages the lrWGS data to study csTL. There are concerns that need to be addressed for this to be considered a useful guide to researchers interested in the expected performance of csTL estimation using lrWGS as concluded by the authors. The key findings largely recapitulate prior studies with regards to variability around csTL and there is limited new insight gained.

One major concern is that srWGS-Telseq is used as a benchmarking approach, and is used to discard the ONT csTL calls but it is well documented that srWGS telseq TL calls are fraught with technical issues that need to be carefully addressed prior to their use in analysis. This seems to be not considered at all, and any comparison to Telseq TL is not robust without that. There appears to be a major missed opportunity to directly do head-to-head comparisons within AoU where there is PacBio and ONT data on an overlapping set of individuals. I would recommend a thorough evaluation of the technical artifacts from batches as it pertains to the srWGS first if the authors want to use the srWGS telseq calls as benchmarking, and even there I would urge some caution.

Another major concern is that analysis presented in Fig 5 suggests strongly that the inclusion of 'batch' as a fixed effect may not be appropriate. There appears to be strong confounding (also noted by the authors) by batch and ancestry possibly even BMI. All downstream analysis subsequent to Fig 5 appears to just use batch as a fixed effect as presented in the corresponding methods sections. In fact this raises concerns about the statements made around ancestry in lines 284-288 but also for all results presented after line 284. Similar concerns with respect to the clinical outcomes.

The argument to exclude chromosome arms to facilitate comparison to Karimian is not well-supported. The argument is not convincing given the data presented in Fig 2D. Why would 15q be included but Xp not? In the AoU data, 15q is as poorly captured as 15p. If the purpose is to facilitate comparisons with the one prior effort, then that would be considered a separate subset analysis. However for a robust effort on AoU, the choices need to be driven by observations in these data directly. Especially as the only place for this comparison is Fig 4C.

There are some numbers that don't reconcile between the text and Table 1. For example: there are 46 in mid-pass HA UW in table 1, but the range of batches is given as 50-971. What is the "*" in table 1? Table would count 11 batches, but text states 10 batches? Also suggest some formality around the term batch which starts at the most granular level of platform-depth-center and then collapses to just platform-depth later in the results.

For the statements around : the correlation between down-sampled and high-coverage csTL estimates remained high across all coverages assessed ($R > 0.75$, $p < 2 \times 10^{-16}$) (Figure S4C - G). : how was this captured when there are fewer arms with cdTL in lower coverage? Was this correlation only limited to those where it is captured by the lowest down sampling? That seems unfair and the correlation should be presented in a way that is clear about this as if this is what was done it fails to incorporate the loss of calls where the correlation would then be 0 - i.e. just not called at all.

For Fig 3 to be fully interpretable: what is the order in panel C are they the same in all, would be more informative to see it ordered same as B. B should be presented once ONT high is removed. Once ONT is removed, is there a relationship between correlation and % of arms called?

The supplementary table titles need much more comprehensive legends.

Reviewer #2

(Remarks to the Author)

In their paper "Determinants of chromosome-specific telomere lengths among 2,573 All of Us participants", Niyati Jain and colleagues describe their results with computational methods to measure chromosome-specific TL (csTL) from long-read whole-genome sequencing (lrWGS) data of 2500 individuals. They confirm findings from several previous studies showing telomere length differences between individuals and between chromosome arms and they observed that the correlation between age and average telomere length was stronger for chromosome arms with longer telomere repeat arrays. The authors provide some lrWGS quality metrics that impact csTL estimation and claim that csTL estimates can be used to estimate disease associations, but I could not find the data supporting this claim. Several interesting findings are reported including differences between ONT and PacBio sequencing platforms, the need to sequence deep (>30X) to get meaningful data and failure to get TL estimates for several arms, including the short arms of the acrocentric autosomes. This data is worth widespread dissemination as many people are currently trying to use long read sequencing technologies to develop telomere length data on specific chromosome arms. In my opinion the paper can be much improved by addressing a few major points.

Major points

1. It is surprising that no attempt was made to haplotype the chromosome-arm specific telomere length estimates. Earlier studies using both Q-FISH (ref 43) and ONT sequencing (ref 40) showed remarkable differences between some haplotype-specific chromosome arm telomere length. Were such differences not observed in the current study? Was no attempt made to separate haplotypes based on SNPs in subtelomeric DNA?
2. It is suggested that frequent recombination between pseudo-homologous regions at the short arms of acrocentric chromosomes may have prevented reliable and unambiguous assignment of telomeres to these arms. Could the authors provide data and a model to support this statement?
3. The shorter telomere length data obtained with ONT suggest that sequencing telomeric DNA is problematic on this platform. Can the authors speculate about possible technical explanations? Is it possible that slight modifications of the sequencing protocol would have shown different results? If so, this should be mentioned.
4. Is it possible that very long telomeres are just harder to sequence? Technical limitations may cooperate with biological factors to explain the loss of long telomere tracts at individual telomeres with age. This possibility should be discussed.
5. The authors speculate that "as the utilization of long-read sequencing technologies increases, lrWGS will likely become a common approach for measuring genetic variation in human cohorts". While this may be true for cohort studies, it is important to mention that clinically validated and useful average telomere length measurements can be made using blood samples from single patients (PMID: 29463756). This observation should be contrasted with the last sentence in the abstract "Larger studies of csTL are needed to advance our understanding of telomeres in aging and disease". Some comments about cohort studies and accuracy of individual measurements will no doubt help aspiring telomere scientists.

Minor points

Many references refer not to original work but reviews and summaries.

Version 1:

Reviewer comments:

Reviewer #1

(Remarks to the Author)

The authors have appropriately addressed all my prior comments.

Reviewer #2

(Remarks to the Author)

My major concerns have been addressed

REVIEWER COMMENTS

Reviewer #1 (Remarks to the Author): I find this work to be interesting and an exciting application of the AoU genomic data as it leverages the lrWGS data to study csTL. There are concerns that need to be addressed for this to be considered a useful guide to researchers interested in the expected performance of csTL estimation using lrWGS as concluded by the authors. The key findings largely recapitulate prior studies with regards to variability around csTL and there is limited new insight gained.

1. One major concern is that srWGS-TelSeq is used as a benchmarking approach and is used to discard the ONT csTL calls but it is well documented that srWGS telseq TL calls are fraught with technical issues that need to be carefully addressed prior to their use in analysis. This seems to be not considered at all, and any comparison to Telseq TL is not robust without that. There appears to be a major missed opportunity to directly do head-to-head comparisons within AoU where there is PacBio and ONT data on an overlapping set of individuals. I would recommend a thorough evaluation of the technical artifacts from batches as it pertains to the srWGS first if the authors want to use the srWGS telseq calls as benchmarking, and even there I would urge some caution.

We thank the reviewer for raising these points. Regarding the use of TelSeq as a benchmarking approach, our decision to exclude ONT data was not based solely on TelSeq comparisons. Rather, we considered the known limitations of Telogator2 with ONT data processed with older base callers (used in All of Us) as well as the weak to absent associations of ONT-based csTL estimates with age (Figure S6). We have reframed the Results section accordingly to clarify that the decision to exclude ONT data was not based solely on TelSeq.

In addition, to address potential batch-related technical artifacts affecting TelSeq estimates, we followed guidance from prior large scale studies that utilized TelSeq, including the NHLBI Trans-Omics for Precision Medicine (Taub et al, Cell Genomics 2022), UK Biobank (Burren et al., Nature Genetics 2024), and All of Us (Nakao et al, preprint 2025). Specifically, we modified our TelSeq analysis by increasing the k parameter to 12 — minimum threshold for the number of telomere repeats required for a read to be considered telomeric — and adjusted for technical confounders using principal component analysis of sequencing depth data. These updates are now described in the Methods section, and the relevant figures (Figure 3, Figure S10) have been revised to show the adjusted TelSeq-TL.

Finally, we agree that head-to-head comparison of PacBio and ONT data would be informative and appreciate the reviewer's suggestion. However, given the small subset of individuals that have data using both platforms (n = 41), such comparisons are limited. We do note in the Methods section under the sub-header "Estimation of csTLs using Telogator2" that we preferred PacBio data for this overlapping subset as it yielded more csTL estimates per individual.

The second and third paragraph under the "Benchmarking csTLs against TelSeq-derived average TL" sub-header of the Results Section now reads: "To further validate csTL estimates generated by Telogator2, we compared them with estimates from TelSeq, a widely used tool for estimating average TL from srWGS data. To account for technical sources of variability in TelSeq-derived TL, we regressed out batch principal components (PCs) derived from sequencing depth data as described previously (6,8,51), optimally selecting the number of PCs (n=30) that maximized the age correlation. In analyses including all participants, adjusted TelSeq-TL was positively correlated with csTL across all chromosome arms ($P < 0.01$) (Figure 3B).

Across the five aggregated sequencing batches, the Sequel IIe and Revio batches showed clear positive correlations with adjusted TelSeq-TL across nearly all chromosome arms ($P < 0.01$). In contrast, ONT-based csTL estimates showed the least consistent correlation with adjusted TelSeq-TL, with most chromosome arms showing no significant association ($P > 0.05$) (**Figure 3C**). Given known limitations of ONT data processed with older base callers (e.g., Guppy used in AoU) for estimating csTLs using Telogator2, we excluded ONT-derived batches from downstream analyses to avoid introducing bias. Moreover, the weaker correlations observed for ONT data relative to the smaller PacBio batches further highlights the potential limited reliability of Telogator2 csTL estimates derived from ONT data in AoU (see Discussion)."

The paragraphs under the "Estimation of average TL using TelSeq" sub-header of the Methods section now reads: "We used a publicly available Docker container (jweinstk/telseq) to facilitate processing of the srWGS data to generate average TL estimates. The container runs a pipeline that takes in a list of CRAM files, converts them to BAM format using SAMtools (75), and runs TelSeq (32) for average TL estimation. It also integrates with workflow management system Cromwell (73) for parallel processing. The TelSeq parameter k was set to 12 (read length 151bp) in line with previous studies (6,8,51). We ran this pipeline for individuals with lrWGS data ($n=2,740$).

Additionally, to account for technical sources of variability, we adjusted our TelSeq-TL with sequencing depth data, as described previously (6,8,51). Briefly, we ran Mosdepth (76) on srWGS CRAM file in bins of 1000 bp across the genome. Next, we generated PCs from the sequencing depth data using NGS-PCA (<https://github.com/PankratzLab/NGS-PCA>), excluding regions with low mappability and a curated list of known problematic structural variants. We calculated batch-adjusted estimates by regressing out the top 30 batch PCs and used the residuals as the adjusted TelSeq-TL. To validate the reliability of our TelSeq-TL estimates, we confirmed that both the unadjusted TelSeq-TL ($R = -0.38$, $P < 2 \times 10^{-16}$) and adjusted TelSeq-TL ($R = -0.53$, $P < 2 \times 10^{-16}$) showed the expected negative correlation with age, with a stronger correlation after PC adjustment (**Figure S10 A and B**)."

2. Another major concern is that analysis presented in Fig 5 suggests strongly that the inclusion of 'batch' as a fixed effect may not be appropriate. There appears to be strong confounding (also noted by the authors) by batch and ancestry possibly even BMI. All downstream analysis subsequent to Fig 5 appears to just use batch as a fixed effect as presented in the corresponding methods sections. In fact, this raises concerns about the statements made around ancestry in lines 284-288 but also for all results presented after line 284. Similar concerns with respect to the clinical outcomes.

We agree that confounding by batch is a concern. Batch adjustment is critical for the appropriate interpretation of associations observed in this dataset. For some variables, such as ancestry, the batching may align so closely with ancestry groups that the needed batch adjustment may attenuate/obscure true associations between ancestry and csTL. This is a limitation of the dataset itself that cannot be overcome through statistical adjustments.

To address concerns about modeling batch as a fixed effect, we compare three batch-correction strategies—fixed effect, random effect, and pre-adjustment—in both the overall and chromosome-arm specific analyses. The statistical significance (P-values), direction and relative magnitude of effect sizes for ancestry, BMI, and other covariates were generally consistent, suggesting that our findings, including those related to clinical outcomes, are robust to batch modeling approach.

The fourth paragraph under the "Estimation of variances in csTL explained by covariates" sub-header of the Methods section reads: "Given that batch is correlated with ancestry and BMI, we compared three batch-correction approaches, modeling batch (i) as a fixed effect, (ii) as a random effect, and (iii) by pre-

adjusting TL for batch prior to analysis. The statistical significance (P-values), the direction and magnitude of effect sizes for each covariate were generally consistent across the three approaches. We present results from models with batch included as a fixed-effect (allowing it to compete with ancestry and BMI) and where TL is modeled as function of batch, yielding TL residuals (isolating variation explained by biologic sources). Results from batch modeled as a random effect is included as a sensitivity analysis (**Table S3**)."

The paragraph under the "Chromosome-arm specific analysis of TL" sub-header of the Methods section now contains the following sentence: "Similar to the overall analysis, we compared three batch-correction approaches. We present results from analyses with batch modeled as (1) a fixed-effect, (2) where TL is modeled as a function of batch and (3) a random effect (**Tables S4 – S7**)."

The third paragraph under the "Determinants of variation in TL" sub-header of the Results section now contains the following sentence: "Batch was strongly correlated with ancestry — with some batches comprised entirely of participants from a single ancestry — and BMI; therefore, even with careful batch adjustment, we were likely limited in our ability to accurately assess the contribution of ancestry and BMI to variation in csTL due to confounding by batch (**Figure 5C and Tables S1-3**)."

3. The argument to exclude chromosome arms to facilitate comparison to Karimian is not well-supported. The argument is not convincing given the data presented in Fig 2D. Why would 15q be included but Xp not? In the AoU data, 15q is as poorly captured as 15p. If the purpose is to facilitate comparisons with the one prior effort, then that would be considered a separate subset analysis. However, for a robust effort on AoU, the choices need to be driven by observations in these data directly. Especially as the only place for this comparison is Fig 4C.

The decision to exclude these chromosome arms was made *a priori* based on their known biology and prior observations in Karimian et al. (2024), and we have now improved our explanation of this decision in the text (described below). As such, while 15q is relatively poorly captured in AoU and Xp performs better in AoU, our exclusion criteria were guided by biological rationale rather than dataset-specific performance. Excluding additional arms such as 15q, for which no specific biological limitation is known, is expected to have minimal impact on downstream results or interpretations.

The last paragraph under the "Long-read sequencing metrics and Telogator2 performance" sub-header of the results section now reads: "Lastly, we evaluated Telogator2's performance across individual chromosome arms and found that csTL estimates were frequently missing for several arms, even in high-coverage datasets, with >30% of samples lacking an estimate for some (**Figure 2D**). The pseudo-homologous regions at the short arms of acrocentric chromosomes (13p, 14p, 15p, 21p, 22p) and the pseudo-autosomal regions at the ends of the X and Y chromosomes facilitate frequent recombination between non-homologous chromosomes (48–50). Such recombination is known to prevent reliable and unambiguous assignment of telomeres to these arms, as demonstrated by Karimian et al. (2024). The authors found that no reads from these arms in the Pangenome dataset mapped to the same chromosome end across three haploid reference assemblies. Therefore, we excluded these arms from further TL analyses. This exclusion resulted in the loss of one participant, leaving 2,572 participants for downstream analyses. Consistent with prior findings, the short arms of acrocentric autosomes (e.g., 13p, 14p, 15p) and the arms of sex chromosomes (e.g., Xp, Yq) were among the underperforming arms across all 5 AoU aggregated batches (defined by platform and coverage) and in the 36 high-coverage (>30×) HPRC samples sequenced using PacBio Revio (**Figure 2D**)."

4. There are some numbers that don't reconcile between the text and Table 1. For example: there are 46 in mid-pass HA UW in table 1, but the range of batches is given as 50-971. What is the "*" in table 1? Table would count 11 batches, but text states 10 batches? Also suggest some formality around the term batch which starts at the most granular level of platform-depth-center and then collapses to just platform-depth later in the results.

Thank you for pointing out these issues. We have corrected numbers in the text to reflect what's shown in Table 1.

The corresponding sentence under the "Characteristics of All of Us participants" sub-header of the Results section now reads: "These technical differences defined 11 unique batches, with batch sizes ranging from 46 participants (center: HA, platform: Revio, coverage: mid-pass) to 971 participants (center: HA, platform: Sequel IIe, coverage: mid-pass)."

The "*" in Table 1 indicates that the batch defined as center: HA, platform: Sequel IIe, coverage: mid-pass has a minimum coverage threshold of 8x, as opposed to 12x for other mid-pass batches. We now clarify as described below.

Table 1 legend now reads: "Metrics in table are shown as mean (SD) or count (%). Participants' ancestry was based on ancestry prediction information provided by All of Us. To protect participant privacy and in accordance with All of Us data and statistics dissemination policy, ancestry groups with fewer than 20 individuals are not shown. Empty categories may include zero or low-count values. For reporting purposes, we combined counts of sex, smoking status and ancestry across centers that used the same sequencing platform and coverage threshold. Centers marked with an asterisk (*) under the mid-pass (12x or 8x) column represent samples sequenced at a minimum of 8x coverage, whereas unmarked centers represent samples sequenced at a minimum of 12x coverage."

We now use the term "batch" when describing the 11 platform-depth-center categories and "aggregated batch" when describing the 5 platform-depth categories.

Figure 2, Panel D legend now reads: "Scatterplot showing the proportion of samples with a csTL estimate for each chromosome arm (ordered by the proportions observed in the Human Pangenome Reference Consortium [HPRC] dataset). The color and shape of the dot denote the data source (HPRC or one of the five All of Us aggregated batches defined by sequencing platform and minimum coverage threshold). Grey bars and red x axis labels denote acrocentric and sex chromosomes. chrYp is not displayed as estimates were only available for five All of Us samples and in none of the HPRC samples."

The sixth paragraph under the "Long-read sequencing metrics and Telogator2 performance" sub-header of the Results section now contains the following sentence: "Consistent with prior findings, the short arms of acrocentric autosomes (e.g., 13p, 14p, 15p) and the arms of sex chromosomes (e.g., Xp, Yq) were among the underperforming arms across all 5 AoU aggregated batches (defined by platform and coverage) and in the 36 high-coverage (>30x) HPRC samples sequenced using PacBio Revio (**Figure 2D**)."

The second paragraph under the "AoU Covariates" sub-header of the Methods section now contains the following sentence: "For some downstream analyses and visualizations, results were aggregated across centers, yielding five aggregated batches defined based on the sequencing platform and minimum coverage threshold."

5. For the statements around: the correlation between down-sampled and high-coverage csTL estimates remained high across all coverages assessed ($R > 0.75$, $p < 2 \times 10^{-16}$) (Figure S4C - G): how was this captured when there are fewer arms with csTL in lower coverage? Was this correlation only limited to those where it is captured by the lowest down sampling? That seems unfair and the correlation should be presented in a way that is clear about this as if this is what was done it fails to incorporate the loss of calls where the correlation would then be 0 - i.e. just not called at all.

The reviewer is correct. The correlation coefficient was calculated based on arms with csTL estimates present in both the high-coverage and down-sampled data. This means that correlation coefficients do not reflect the loss of calls. The relationship between coverage and the number of arms called is displayed in Figure S4A. We now clarify that the scatterplots and correlation coefficients shown in Figure S4 are meant to reflect the consistency between csTLs that are successfully estimated. The high correlations observed remains informative, as it highlights that when a csTL estimate is available in both the high-coverage and down-sampled data, they strongly correlate, supporting validity of csTL estimation from lower coverage data, even when the total number of csTLs called per sample is small due to low coverage.

The fifth paragraph under the sub-header “Long-read sequencing metrics and Telogator2 performance” of the Results section now contains the following sentence: “However, the correlation between csTL estimates present in both down-sampled and high-coverage data remained high across all comparisons ($R > 0.75$, $p < 2 \times 10^{-16}$). Because lower coverage datasets produce fewer csTL estimates, the correlations estimated for low coverage datasets are based on few observations (Figure S4C - G).”

The paragraph under the sub-header “Long-read WGS metrics and Telogator2 performance” of the Methods section now contains the following sentence: “Down sampling was performed using the software Sambamba (78), and we assessed the correlation (Pearson’s) and absolute mean error in Telogator2 csTL estimates between high coverage and down sampled data, considering only csTL estimates that were present in both.”

In Figure S4C - G, we now include the number of csTL observations used for calculating the correlation between high-coverage and down sampled data. The Figure S4C-G figure legend now includes the following sentence: “Pearson’s correlation coefficient (R), the P-value and the number of csTL observations (n) used to compute the correlation are displayed.”

6. For Fig 3 to be fully interpretable: what is the order in panel C are they the same in all, would be more informative to see it ordered same as B. B should be presented once ONT high is removed. Once ONT is removed, is there a relationship between correlation and % of arms called?

The ordering in panel C differs across the five aggregated batches (defined by platform and coverage threshold) and therefore does not match the ordering shown in panel B. In panel C, chromosome arms are ordered within each aggregated batch based on the magnitude of their correlation with adjusted TelSeq-TL. The purpose of this panel is to illustrate overall differences in correlation patterns across aggregated batches, rather than comparing TelSeq correlations at individual chromosome arms. Thus, we prefer not to reorder panel C, as it would be less effective at illustrating the overall differences across batches.

Following the reviewer's suggestion, we now present panel B without ONT as a supplementary figure (**Figure S11**).

Finally, we assessed whether after removing ONT samples, there is a relationship between correlation strength and the sample size for each chromosome arm (number of individuals with csTL estimates). We found no evidence of such an association, either when ONT samples were included ($R = 0.15$, $P = 0.36$) or excluded ($R = 0.045$, $P = 0.79$).

7. The supplementary table titles need much more comprehensive legends.

We have now updated all the supplementary tables with footnotes and revised titles.

Reviewer #2 (Remarks to the Author):

In their paper "Determinants of chromosome-specific telomere lengths among 2,573 All of Us participants", Niyati Jain and colleagues describe their results with computational methods to measure chromosome-specific TL (csTL) from long-read whole-genome sequencing (lrWGS) data of 2500 individuals. They confirm findings from several previous studies showing telomere length differences between individuals and between chromosome arms and they observed that the correlation between age and average telomere length was stronger for chromosome arms with longer telomere repeat arrays. The authors provide some lrWGS quality metrics that impact csTL estimation and claim that csTL estimates can be used to estimate disease associations, but I could not find the data supporting this claim. Several interesting findings are reported including differences between ONT and PacBio sequencing platforms, the need to sequence deep (>30X) to get meaningful data and failure to get TL estimates for several arms, including the short arms of the acrocentric autosomes. This data is worth widespread dissemination as many people are currently trying to use long read sequencing technologies to develop telomere length data on specific chromosome arms. In my opinion the paper can be much improved by addressing a few major points.

The data supporting the claim that csTL estimates can be used to estimate disease associations can be found in Figure S5 and Supplementary Table S8. While we don't identify statistically significant disease associations — likely due to limited power — our goal was to highlight that csTL estimation enables assessment of novel TL metrics, such as the shortest telomere or associations at the individual chromosome arm level.

Major points

1. It is surprising that no attempt was made to haplotype the chromosome-arm specific telomere length estimates. Earlier studies using both Q-FISH (ref 43) and ONT sequencing (ref 40) showed remarkable differences between some haplotype-specific chromosome arm telomere length. Were such differences not observed in the current study? Was no attempt made to separate haplotypes based on SNPs in subtelomeric DNA?

We thank the reviewer for raising this point. Although Telogator2 can estimate csTLs at the haplotype level (i.e., two csTL estimates per chromosome arm), in the AoU dataset we observe that for a given arm, the majority of individuals have only a single csTL estimate (**Figure S9**). A smaller proportion of individuals have exactly two or in some cases even more than two csTL estimates for a given arm. This pattern is likely driven by variability in telomeric coverage, which affects how many csTL estimates can

be reliably estimated per sample, with lower coverage samples yielding fewer csTL estimates overall, as shown in **Figure 2B** and **2C**. Because of this variability, robust analysis of haplotype-specific differences in csTL were not feasible. Therefore, in the Methods section (fourth paragraph under sub-header “Estimation of csTLs using Telogator2”) we describe that when two or more estimates were available for a given arm, we averaged them, and when only one estimate was available, that single estimate was used as the csTL estimate.

We have modified Figure S9 to more clearly show this pattern, and we have added relevant text to both the Results and Discussion section.

The Figure S9 legend now reads: “Distribution of the number of TL estimates per chromosome arm across individuals. The x-axis shows each chromosome arm, and the y-axis indicates the number of individuals. Bars represent a stacked count of individuals with 1, 2, or >2 csTL estimates for that arm.”

The fourth paragraph under the sub-header “Long-read sequencing metrics and Telogator2 performance” of the Results section now contains the following sentence: “Additionally, we observed that for a given arm, majority of individuals had only a single csTL estimate (**Figure S9**), thus, motivating our approach of reporting the average csTL when 2 or more estimates were available or the single estimate when only one was present.”

The second paragraph under of the Discussion section now contains the following sentence: “Additionally, because most individuals have a single csTL estimate for a given arm —likely driven by the large subset of low telomeric coverage samples in AoU (**Figure 2C**) — our ability to assess haplotype-specific differences in csTLs was limited, highlighting the need for future studies with improved telomeric coverage to allow for this type of analysis.”

2. It is suggested that frequent recombination between pseudo-homologous regions at the short arms of acrocentric chromosomes may have prevented reliable and unambiguous assignment of telomeres to these arms. Could the authors provide data and a model to support this statement

The decision to exclude these chromosome arms was made *a priori* based on their known biology and observations in Karimian et al. (2024). Specifically, the recombination activity which prevents reliable assignment of telomeres was directly quantified by Karimian et al. 2024 using the Human Pangenome Reference Consortium (HPRC) data. We now describe their findings to justify exclusion of these arms.

The last paragraph under the “Long-read sequencing metrics and Telogator2 performance” sub-header of the results section now reads: “Lastly, we evaluated Telogator2’s performance across individual chromosome arms and found that csTL estimates were frequently missing for several arms, even in high-coverage datasets, with >30% of samples lacking an estimate for some (**Figure 2D**). The pseudo-homologous regions at the short arms of acrocentric chromosomes (13p, 14p, 15p, 21p, 22p) and the pseudo-autosomal regions at the ends of the X and Y chromosomes facilitate frequent recombination between non-homologous chromosomes (48–50). Such recombination is known to prevent reliable and unambiguous assignment of telomeres to these arms, as demonstrated by Karimian et al. (2024). The authors found that no reads from these arms in the Pangenome dataset mapped to the same chromosome end across three haploid reference assemblies. Therefore, we excluded these arms from further TL analyses. This exclusion resulted in the loss of one participant, leaving 2,572 participants for downstream analyses. Consistent with prior findings, the short arms of acrocentric autosomes (e.g., 13p, 14p, 15p) and the arms of sex chromosomes (e.g., Xp, Yq) were among the underperforming arms across

all 5 AoU aggregated batches (defined by platform and coverage) and in the 36 high-coverage (>30×) HPRC samples sequenced using PacBio Revio (**Figure 2D**)."

3. The shorter telomere length data obtained with ONT suggest that sequencing telomeric DNA is problematic on this platform. Can the authors speculate about possible technical explanations? Is it possible that slight modifications of the sequencing protocol would have shown different results? If so, this should be mentioned.

While we cannot offer a definitive technical explanation, we speculate that ONT-specific biases related to base-calling errors could contribute to shorter TL estimates, as well as the weaker associations with age and TelSeq-TL observed for the ONT csTL estimates. It is also possible that DNA extraction and library preparation methods contribute to the observed variability. We have expanded the discussion to include these technical explanations.

The third paragraph of the Discussion section now reads: "Despite sufficient coverage and read lengths, ONT batches were less reliable for csTL estimation, producing systematically shorter csTL estimates and weaker correlations with TelSeq-TL (**Figure 3C**) and age (**Figure S6**) compared to PacBio batches (Revio and Sequel IIe). This reduced reliability could be due to ONT's higher base-calling error rates in repetitive regions. For this reason, the Telogator2 developers noted limitations when analyzing telomeres using older base callers like Guppy (53, 55). Consistently, a recent study showed that ONT reads exhibit systematic miscalls within telomeric repeat regions (e.g., TTAGGG), a phenomenon not observed in PacBio sequencing reads (54). It is therefore plausible that ONT-specific biases lead to inaccurate identification of telomere–TVR boundaries such that miscalled "telomeric" repeats are not recognized as part of telomeric regions, resulting in underestimation of csTLs. The AoU ONT data include reads processed with both Guppy and the newer Dorado base caller, which provides improved accuracy; however, the available metadata did not allow us to distinguish between them. Additionally, because raw reads were unavailable, we could not re-call bases using newer or telomere-trained models (55). Beyond base-calling errors, DNA extraction and library preparation protocols could also influence estimation, though future studies are needed to clarify their effects. Together, these findings highlight ongoing technical challenges in using AoU IrWGS data for csTL estimation."

4. Is it possible that very long telomeres are just harder to sequence? Technical limitations may cooperate with biological factors to explain the loss of long telomere tracts at individual telomeres with age. This possibility should be discussed.

We agree that very long telomeres can be more difficult to sequence, and although our long-read PacBio data (~20 kb mean read length) should capture most telomeres, very long telomeres are more likely underestimated. Regarding the reviewer's suggestion that technical limitations could interact with biological factors, we acknowledge that both could contribute to the observed age-related patterns in our data. However, our understanding is that the underestimation of very long telomeres—likely more common in younger individuals—would tend to attenuate associations between age and TL.

The eighth paragraph of the Discussion section now contains the following sentence: "Although our read lengths (~20 kb for PacBio batches) should capture most telomeres, very long telomeres—likely more common in younger individuals—may be underestimated, which could lead to some attenuation of the age-related associations."

5. The authors speculate that "as the utilization of long-read sequencing technologies increases, IrWGS

will likely become a common approach for measuring genetic variation in human cohorts". While this may be true for cohort studies, it is important to mention that clinically validated and useful average telomere length measurements can be made using blood samples from single patients (PMID: 29463756). This observation should be contrasted with the last sentence in the abstract "Larger studies of csTL are needed to advance our understanding of telomeres in aging and disease". Some comments about cohort studies and accuracy of individual measurements will no doubt help aspiring telomere scientists.

The last paragraph of Discussion section now contains the following sentences: "Given the suggestive trends observed in our analysis, and evidence from prior studies, we propose that incorporating csTL estimates into large research studies of diverse human population cohorts, including assessment of TL distributions and extreme values (both short and long), has the potential to provide important insights into the variation of csTLs across individuals and the roles of csTLs in human aging and disease. At the same time, it is important to recognize that high-quality mean TL measurements from single patients using accurate, clinically-validated methods show clear utility for diagnostics and treatment decisions (68). While mean TL will continue to be an important measure in improving our understanding of human aging and disease biology, complementing it with large scale csTL analyses will further bridge research findings with clinical applications."

The Conclusion section now omits the following sentence: "Larger studies of diverse human cohorts are needed to validate our results and further examine the variation in csTL across individuals and the roles of csTLs in human aging and disease."

Minor points

Many references refer not to original work but reviews and summaries.

We have updated the references such that they refer to the original work rather than reviews and summaries, including the papers describing association of telomere length with stress (reference 11) and with smoking (references 14 and 15).